# Reward-Shaping Control Variates for Off-Policy Evaluation under Sparse Rewards

## Abstract

Off-policy evaluation (OPE) is essential for deploying reinforcement learning in safety-critical settings, yet existing estimators such as importance sampling and doubly robust (DR) often exhibit prohibitively high variance when rewards are sparse. In this work, we introduce Reward-Shaping Control Variates, a new family of unbiased estimators that leverage potential-based reward shaping to construct additional zero-mean control variates. We prove that shaped estimators always yields valid variance reduction, and that combining shaping-based and Q-based control variates strictly expands the variance-reduction subspace beyond DR and its minimax variant MRDR. Empirically, we provide a systematic regime map across synthetic chains, a cancer simulator, and an ICU-sepsis benchmark showing that shaping-based OPE consistently outperforms DR in sparse-reward settings, while a hybrid estimator achieves state-of-the-art performance across sparse, noisy, and misspecified environments. Our results highlight reward shaping as a powerful and interpretable tool for robust OPE, offering both theoretical guarantees and practical improvements in domains where standard estimators fail.

## 1 Introduction

Off-policy evaluation (OPE) is a central problem in reinforcement learning (RL): given data collected by a *behavior policy*, the goal is to estimate the performance of a different *evaluation policy* without deploying it in the environment. Reliable OPE is crucial in high-stakes applications such as healthcare, education, and recommendation systems, where deploying untested policies can have significant costs. Standard OPE methods fall broadly into two families: importance sampling (IS) (Precup et al., 2000; Jiang & Li, 2016), which reweights trajectories to mimic the evaluation policy, and doubly robust (DR) methods (Dudík et al., 2011; Thomas & Brunskill, 2016), which combine reweighting with regression-based reward prediction. These estimators are well studied and can perform effectively when rewards are dense and trajectories contain frequent signals. However, many real-world settings are characterized by *sparse rewards*, where meaningful feedback occurs only rarely or at the end of a long horizon. In such cases, existing estimators become unreliable.

To see why, consider a simple illustrative example. Imagine a maze with 100 decision steps, where the agent receives a reward of +1 only if it reaches the goal at the very end. Suppose the behavior policy succeeds 1% of the time, while the evaluation policy succeeds 10% of the time. In this setting, nearly all trajectories generated by the behavior policy contain zero reward, and only the rare successful trajectories provide useful information. Importance sampling collapses under this regime: because nonzero weights arise only when success is observed, the variance of the estimate grows prohibitively large, requiring thousands of samples to stabilize. Doubly robust methods also fail: the regression model must extrapolate the outcome of reaching the goal from extremely scarce reward observations, leading to high bias.

Marginalized estimators, such as DualDICE (Nachum et al., 2019) and GenDICE (Zhang et al., 2020a), reduce variance by directly estimating state(-action) distribution ratios rather than trajectory weights. While effective in moderately sparse domains, they remain brittle in the extreme setting described above. The reason is that these estimators still depend entirely on reward support: when nearly all observed rewards are zero, the optimization problem becomes ill-posed, and the learned distribution ratios are uninformative for evaluation. In practice, this yields estimates with low variance but high bias—the estimator stabilizes around a misleading value be-

cause the reward signal is too scarce to propagate through the state space. Moreover, when the evaluation policy succeeds in regions where the behavior policy almost never collects reward, marginalized estimators inherit the same extrapolation gap as DR methods (Uehara et al., 2020).

This simple example illustrates a broader problem: *Standard OPE estimators are fundamentally unreliable in sparse-reward environments, precisely the regimes where reliable evaluation is most needed.* This limitation underscores the need for new approaches that can extract signal even when rewards are infrequent or noisy.

In this paper, we address this challenge by introducing *reward-shaping control variates (RSCV)*, a new class of estimators that reduce variance without introducing bias. Our contributions are as follows: We introduce a theoretical framework showing how potential-based reward shaping can be leveraged a control variate in OPE. We prove that the resulting estimators remain unbiased while achieving strict variance reduction compared to standard IS, DR, and marginalized estimators. We analyse sparse and noisy reward regimes, identifying conditions under which shaping control variates provides significant improvements in performance. We demonstrate with a suite of experiments on tabular MDPs, a cancer simulator (Ribba et al., 2012) and an ICU-Sepsis benchmark (Choudhary et al., 2024), that shaped estimators consistently outperform existing OPE baselines in sparse- and noisy-reward environments. We further provide practical guidance on constructing shaping functions for real-world applications, highlighting the trade-off between variance reduction and robustness. Together, these contributions establish reward-shaping control variates as a principled and practical solution for robust off-policy evaluation in settings where standard estimators fail.

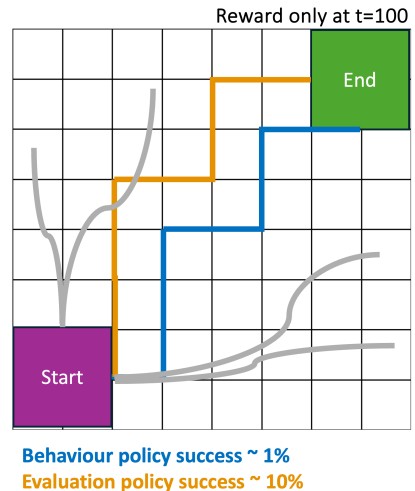

Figure 1: Sparse-reward maze example. Most behaviour rollouts (grey) fail and receive 0 reward; only 1% (blue) succeed with +1. The evaluation policy succeeds 10%, but off-policy estimates (orange) require thousands of samples to stabilize due to the scarcity of reward in behavior data.

## 2 RELATED WORK

**IS, DR and Marginalized Estimators.** A well-established body of work in OPE includes importance sampling (IS) (Precup et al., 2000; Jiang & Li, 2016) and doubly robust (DR) methods (Dudík et al., 2011; Thomas & Brunskill, 2016). IS reweights trajectories to match the evaluation policy but suffers exponential variance in sparse-reward settings, as rare successes dominate the weights. DR reduces variance by combining IS with a learned reward model, but incurs bias when extrapolating from scarce rewards. In contrast, our method instead applies potential-based reward shaping as a control variate, reducing variance without dense rewards. Marginalized estimators such as DualDICE (Nachum et al., 2019), GenDICE (Zhang et al., 2020a), and VPM (Uehara et al., 2020) estimate stationary state–action distributions rather than trajectory weights, lowering variance but remains biased under extreme sparsity, where optimization of the density ratio becomes ill-conditioned. Our approach injects informative shaped rewards throughout the trajectory, enabling reliable evaluation even when sparse feedback offers little direct signal.

**Model-Based and Representation-Learning Approaches.** Another line of OPE research leverages model-based rollouts (Farahmand et al., 2018; Le et al., 2019) or representation learning (e.g., invariant or confounder-aware embeddings) to mitigate distributional shifts (Hanna et al., 2017; Uehara et al., 2022).These however require accurate dynamics or strong sufficiency assumptions. Our method sidesteps this by reshaping rewards rather than modeling dynamics. Majumdar et al. (2025) incorporate human-interpretable "concepts" to reduce variance while remaining unbiased, assuming concept supervision or the ability to discover reflective abstractions. While also targeting variance reduction, their method depends on concept supervision, whereas ours directly reshapes rewards, enhancing evaluation fidelity even in standard RL settings with known reward functions.

**Reward Shaping in Reinforcement Learning.** Potential-based reward shaping accelerates learning without changing optimal policies (Ng et al., 1999), and has been used to improve sample efficiency in online RL (Devlin & Kudenko, 2012; Grzes, 2016). Prior OPE work considered fixed shaping-based control variates (Parbhoo et al., 2020), but only in limited settings. Unlike Parbhoo et al. (2020), we explicitly learn zero-mean control variates for variance reduction, prove unbiasedness and variance reduction for varying strengths of the control variates and also provide an algorithm that jointly optimizes the potential function and its strength to guarantee variance reduction. This generalization strictly subsumes Parbhoo et al. (2020) and achieves state-of-the-art performance in sparse, noisy, and misspecified environments.

## 3 Preliminaries and Notation

**Markov Decision Processes.** A Markov decision process (MDP) is a tuple $M = (\mathcal{S}, \mathcal{A}, P, \gamma, R)$, where $\mathcal{S}$ is the state space, $\mathcal{A}$ the action space, $P(s, a, s')$ the transition distribution from $s$ to $s'$ under action $a$, $R(s, a)$ the reward for $(s, a)$, and discount factor $\gamma$. A stationary policy $\pi : \mathcal{S} \times \mathcal{A} \to [0, 1]$ maps states to action probabilities, with $\pi(a|s)$ the probability of taking $a$ in $s$. A trajectory $\tau = (s_0, a_0, r_0, \ldots, s_T)$ generated by $\pi$ has return $R_{0:T-1}(\tau) = \sum_{t=0}^{T-1} \gamma^t r_t$, where $s_{t+1} \sim P(\cdot|s_t, a_t)$ and $a_t \sim \pi(\cdot|s_t)$. The performance of $\pi$ is $V^\pi = \mathbb{E}_{P_\tau^\pi}[R_{0:T-1}(\tau)]$. The value and action-value functions, $V^\pi(s)$ and $Q^\pi(s, a)$, are the expected returns starting from $s$, or from $(s, a)$ followed by $\pi$, respectively.

**Off-Policy Evaluation.** Given a dataset $\mathcal{D} = \{\tau^{(i)}\}_{i=1}^n$ of trajectories generated by a *behavior policy* $\pi_b$, we wish to estimate the value of a different *evaluation policy* $\pi_e$. A valid estimator $\hat{V}^{\pi_e}$ should minimize the mean squared error (MSE):

$$\text{MSE}(V^{\pi_e}, \hat{V}^{\pi_e}) = \underbrace{\left(\mathbb{E}_{P_\tau^{\pi_b}}[\hat{V}^{\pi_e}] - V^{\pi_e}\right)^2}_{\text{Bias}^2} + \underbrace{\text{Var}(\hat{V}^{\pi_e})}_{\text{Variance}}.$$

where $P_\tau^{\pi_b}$ denotes the distribution of trajectory $\tau$ under behaviour policy $\pi_b$. This decomposition highlights that reducing variance while maintaining unbiasedness is central to designing effective OPE estimators. We adopt the following standard assumptions:

**Assumption 1 (Absolute Continuity).** For all $(s, a) \in \mathcal{S} \times \mathcal{A}$, if $\pi_b(a \mid s) = 0$, then $\pi_e(a \mid s) = 0$.

**Assumption 2 (Single Behavior Policy).** All trajectories in $\mathcal{D}$ are sampled independently under the same behavior policy $\pi_b$.

**Potential-Based Reward Shaping.** Reward shaping is a technique that is used to modify the original reward function using a reward-shaping function $F : \mathcal{S} \times \mathcal{A} \times \mathcal{S} \to \mathbb{R}$ to typically make RL methods converge faster with more instructive feedback. The original MDP $M = (\mathcal{S}, \mathcal{A}, P, \gamma, R)$ is transformed into a *shaped-MDP* $M' = \mathcal{S}, \mathcal{A}, P, \gamma, R' = R + F)$. Reward shaping modifies the original reward signal using an auxiliary shaping function $F : \mathcal{S} \times \mathcal{A} \times \mathcal{S} \to \mathbb{R}$, yielding a new reward function

$$R'(s, a, s') = R(s, a) + F(s, a, s').$$

While arbitrary shaping may alter the optimal policy, *potential-based reward shaping (PBRS)* (Ng et al., 1999) preserves policy invariance.

**Definition 1 (Potential-Based Reward Shaping).** *A shaping function $F$ is potential-based if there exists a potential $\phi : \mathcal{S} \to \mathbb{R}$ such that*

$$F(s, a, s') = \gamma\phi(s') - \phi(s), \quad \forall (s, a, s') \in \mathcal{S} \times \mathcal{A} \times \mathcal{S}.$$

**Theorem 1 (Policy Invariance under PBRS).** *If shaping is potential-based, then for all policies $\pi$, the optimal policy under the shaped MDP coincides with the optimal policy under the original MDP. That is, PBRS modifies value functions by a state-dependent constant shift but leaves the optimal action distribution unchanged.*

This invariance property makes PBRS especially appealing for OPE: it allows us to introduce additional signal into sparse-reward environments while guaranteeing that the evaluation policy's value is estimated consistently with the original reward structure. As the learnt shaping potentials depend only on states, there is no contamination to the policy and the environmental setup, maintaining the fidelity of the task.

## 4 REWARD-SHAPING CONTROL VARIATES

In this section we introduce a new class of *reward-shaping control variates* (RSCV) for variance reduction in off-policy evaluation. The key idea is to construct an additional random variable $C^{\Phi}$ with *zero mean* under the behavior-policy distribution. Such a control variate can be added to any unbiased OPE estimator without altering its expectation, but with the potential to substantially reduce variance. We first formalize the construction of $C^{\Phi}$ and then show how it can be integrated into the per-decision importance sampling (PDIS) estimator. While we present results for PDIS, the same construction can be incorporated into other OPE estimators (e.g., weighted IS, DR, MRDR) with analogous guarantees. See Appendix D for details.

**Definition 2** (**A Reward-Shaping Control Variate**). *Let $\Phi : \mathcal{S} \to \mathbb{R}$ be any measurable potential function such that $\Phi(s_T) = 0$ for all terminal states $s_T$. Define a reward-shaping control variate*

$$C^{\Phi} := \Phi(s_0) + \sum_{t=0}^{T-1} \gamma^t W_t \big( \gamma \Phi(s_{t+1}) - \Phi(s_t) \big),$$

*where $W_t = \prod_{k=0}^{t} \frac{\pi_e(a_k|s_k)}{\pi_b(a_k|s_k)}$ is the cumulative importance weight up to time $t$.*

**Lemma 1** (**Zero-mean property**). *Under Assumption 1 (absolute continuity) and the boundary condition $\Phi(s_T) = 0$, the mean of the control variate under $\pi_b$, $\mathbb{E}_{\pi_b}[C^{\Phi}] = 0$. Thus $C^{\Phi}$ is a valid zero-mean control variate for PDIS return $Y = \sum_{t=0}^{T-1} \gamma^t W_t r_t$.*

**Proof sketch:** For each $t$, by a change of measure using importance weights,

$$\mathbb{E}_{\pi_b} \big[ \gamma^t W_t (\gamma \Phi(s_{t+1}) - \Phi(s_t)) \big] = \mathbb{E}_{\pi_e}[\gamma^{t+1} \Phi(s_{t+1}) - \gamma^t \Phi(s_t)].$$

We provide a detailed version of the proof in Appendix C. There are 3 different timesteps under which the potential function $\Phi$ is defined. First, at the final timestep T, we define $\Phi(S_T) = 0$. As we know the reward outcome at the final timestep, we don't have to shape the reward at states which occur at the final timestep $T$. Second, at the first timestep, we assume $\mathbb{E}^{\pi_b}[C^{\Phi}] = \mathbb{E}^{\pi_e}[C^{\Phi}] = 0$, as in practice we have a notion of the distribution of the initial states. Hence, we don't need to necessarily shape the starts that occur at time 0. Finally, from timesteps 1 to $t-1$, we attempt to learn the shaping function $\Phi$ which actively serves as a reward signal while performing OPE.

**Definition 3** (**Shaped PDIS estimator**). *Given $N$ i.i.d. trajectories, the shaped PDIS estimator augments PDIS with the a reward shaping control variate $C^{\Phi}$:*

$$\widehat{V}_{\mathrm{Shaped-PDIS}}^{\pi_e}(\lambda) = \frac{1}{N} \sum_{n=1}^{N} \Big( \sum_{t=0}^{T-1} \gamma^t W_t r_t + \lambda C^{\Phi,n} \Big),$$

*where $\lambda \in \mathbb{R}$ is a coefficient controlling the weight of the zero-mean control variate applied.*

**Theorem 2** (**Unbiasedness of Shaped PDIS**). *For any $\lambda \in \mathbb{R}$, the Shaped PDIS estimator is unbiased. That is, $\mathbb{E}_{\pi_b} \left[ \widehat{V}_{\mathrm{Shaped-PDIS}}^{\pi_e}(\lambda) \right] = V^{\pi_e}$.*

*Proof.* By linearity of expectation,

$$\mathbb{E}_{\pi_b} \left[ \widehat{V}_{\mathrm{Shaped-PDIS}}^{\pi_e}(\lambda) \right] = \mathbb{E}_{\pi_b} \left[ \widehat{V}_{\mathrm{PDIS}}^{\pi_e} \right] + \lambda \mathbb{E}_{\pi_b} \left[ C^{\Phi} \right] = V^{\pi_e}$$

From IS theory, $\mathbb{E}_{\pi_b}[\widehat{V}_{\mathrm{PDIS}}^{\pi_e}] = V^{\pi_e}$ and by Lemma 1, $\mathbb{E}_{\pi_b}[C^{\Phi}] = 0$. Thus, the estimator is unbiased for any choice of $\lambda$. $\square$

### 4.1 VARIANCE ANALYSIS OF REWARD SHAPING CONTROL VARIATES

In what follows, we characterize the variance of $\widehat{V}_{\mathrm{Shaped-PDIS}}^{\pi_e}(\lambda)$ and discuss the optimal choice of $\lambda$ that minimizes the variance, $\lambda^{\star} = -\frac{\mathrm{Cov}[Y, C^{\Phi}]}{\mathrm{Var}[C^{\Phi}]}$.

**Theorem 3** (**Variance decomposition under discounting**). *For any $\lambda$,*

$$\mathrm{Var}[\widehat{V}_{\mathrm{Shaped-PDIS}}(\lambda)] = \frac{1}{N} \Big( \mathrm{Var}[Y] + 2\lambda \mathrm{Cov}[Y, C^{\Phi}] + \lambda^2 \mathrm{Var}[C^{\Phi}] \Big). \tag{1}$$

*The variance-minimizing coefficient is $\lambda^{\star} = -\frac{\mathrm{Cov}[Y, C^{\Phi}]}{\mathrm{Var}[C^{\Phi}]}$ with minimized variance,*

$$\mathrm{Var}[\widehat{V}_{\mathrm{Shaped-PDIS}}(\lambda^{\star})] = \frac{1}{N} \Big( \mathrm{Var}[Y] - \frac{\mathrm{Cov}[Y, C^{\Phi}]^2}{\mathrm{Var}[C^{\Phi}]} \Big). \tag{2}$$

*Proof.* Expand $\mathrm{Var}[Y + \lambda C^\Phi]$. Minimization follows by completing the square in $\lambda$. $\qquad\square$

Unlike DR and its variants which cannot guarantee strict improvement (variance may even rise if the model of $Q$ is bad), Shaped-PDIS has the following guarantee:

**Corollary 1** (**Variance bound**). *The optimal Shaped PDIS estimator never has larger variance than PDIS*

$$\mathrm{Var}\big[\widehat{V}_{\text{Shaped}-\text{PDIS}}(\lambda^\star)\big] \le \tfrac{1}{N}\mathrm{Var}[Y],$$

*with strict inequality iff* $\mathrm{Cov}[Y, C^\Phi] \ne 0$.

**Case $\lambda = -1$.** Note that when $\lambda = -1$, we recover the SCOPE estimator from Parbhoo et al. (2020). Its variance is $\frac{1}{N}\big(\mathrm{Var}[Y] - 2\mathrm{Cov}[Y, C^\Phi] + \mathrm{Var}[C^\Phi]\big)$, which may be larger or smaller than PDIS depending on $2\,\mathrm{Cov}[Y, C^\Phi] \gtrless \mathrm{Var}[C^\Phi]$.

**Case $\lambda = \lambda^\star$ (projection-optimal).** When $\lambda$ is optimized, Shaped-PDIS consistently improves on PDIS by Corollary 1, with improvement factor $1 - \rho_{Y,C^\Phi}^2$, where $\rho$ is the correlation between $Y$ and shaping control variates $C^\Phi$.

### 4.2 Learning control variates $C^\Phi$ and $\lambda$ for variance guarantees

Shaped-PDIS performance hinges on the control variate $C^\Phi$ and coefficient $\lambda$: variance drops whenever $C^\Phi$ correlates with the IS–weighted returns $Y$. We now learn $C^\Phi$ and $\lambda$ directly from data by parameterizing a state potential $\phi_\beta : \mathcal{S} \to \mathbb{R}$ with a neural network and training it end-to-end (via backpropagation) to maximize explained variance between the induced shaping terms and PDIS returns. This aligns the control variate with the return signal and thus guarantees variance reduction in OPE. The full procedure appears in Algorithm 1.

**Computing control variates $C^\Phi$ using a potential network.** The control variates in RSCV are defined by a potential function $\Phi : \mathcal{S} \to \mathbb{R}$, which assigns a scalar "potential value" to each state. $\Phi$ captures progress toward reward or risk of failure, so that the differences $\gamma\Phi(s_{t+1}) - \Phi(s_t)$ absorb predictable structure in the return. To make this practical, we parameterize $\Phi$ by a function approximator $\Phi_\beta$ (e.g., a linear model or neural network) and optimize its parameters $\beta$ for variance reduction. A detailed explanation on the intuition of potential function is provided in Appendix B. Concretely, for each trajectory $n$ we form the per-trajectory return

$$Y^{(n)} = \sum_{t=0}^{T-1} \gamma^t W_t^{(n)} r_t^{(n)}, \quad W_t^{(n)} = \prod_{i=0}^{t} \frac{\pi_e(a_i^{(n)}|s_i^{(n)})}{\pi_b(a_i^{(n)}|s_i^{(n)})},$$

and the shaping columns

$$C^{(n)}(\beta) = \Big[\Phi_\beta(s_0^{(n)}), \ \gamma^t W_t^{(n)}\big(\gamma\Phi_\beta(s_{t+1}^{(n)}) - \Phi_\beta(s_t^{(n)})\big)\Big]_{t=0}^{T-1}.$$

Stacking over trajectories yields the design matrix $\mathbf{C}(\beta)$ and vector $\mathbf{Y}$. After row centering or de-meaning, we compute covariances

$$\mathbf{\Sigma}_{CC}(\beta) = \tfrac{1}{N}\mathbf{C}_c^\top \mathbf{C}_c + \alpha I, \quad \mathbf{\Sigma}_{CY}(\beta) = \tfrac{1}{N}\mathbf{C}_c^\top \mathbf{Y}_c.$$

Centering ensures that the control variates cannot shift the expectation of the estimator and alter its bias, thus stabilizing optimization. The explained variance objective is then

$$J(\beta) = \mathbf{\Sigma}_{CY}(\beta)^\top \mathbf{\Sigma}_{CC}(\beta)^{-1} \mathbf{\Sigma}_{CY}(\beta). \tag{3}$$

Note that in practice when $N, T$ are large, computing the matrix inversion in Eq.3 becomes intractable. To reduce computational overload, one can resort to conjugate gradients (Shewchuk et al., 1994) (See Appendix E for details).

**Optimizing the control variates $C^\Phi$ for variance reduction.** Maximizing $J(\beta)$ via backpropagation directly minimizes the residual variance of Shaped-PDIS, ensuring that the shaping features are aligned with the noisy IS returns. Unlike standard supervised learning, there are no external labels: the only training signal comes from how well the shaping columns correlate with the return.

---

**Algorithm 1** Shaping-based OPE with Learned Potential $\Phi_\beta$

---

**Require:** Trajectories $\{\tau^n\}$ from $\pi_b$, evaluation policy $\pi_e$, discount $\gamma$, potential model $\Phi_\beta$, folds $K$, ridge $\alpha$.

1: **for** $k = 1, \ldots, K$ **do**
2:     Split data: fit set $\mathcal{I}_{\text{fit}}$, evaluation set $\mathcal{I}_k$.
3:     On $\mathcal{I}_{\text{fit}}$, build $\mathbf{Y}, \mathbf{C}(\beta)$ from IS returns $Y^{(n)}$ and shaping columns $C^{(n)}(\beta)$.
4:     Form covariances $\mathbf{\Sigma}_{CC}, \mathbf{\Sigma}_{CY}$; maximize $J(\beta) = \mathbf{\Sigma}_{CY}^\top \mathbf{\Sigma}_{CC}^{-1} \mathbf{\Sigma}_{CY}$ by backprop.
5:     Compute $\lambda_k = -\mathbf{\Sigma}_{CC}^{-1} \mathbf{\Sigma}_{CY}$ at $\hat{\beta}$.
6:     Evaluate on $\mathcal{I}_k$: $\hat{V}_k = \frac{1}{|\mathcal{I}_k|} \sum_{n \in \mathcal{I}_k} (Y^{(n)} + \lambda_k^\top C^{(n)}(\hat{\beta}))$.
7: **end for**
8: **return** $\hat{V}_{\text{RS-CV}}^{\pi_e} = \frac{1}{K} \sum_k \hat{V}_k$.

---

**Cross-fitting and the final estimator.**    To prevent overfitting, we apply cross-fitting: the dataset is partitioned into $K$ folds, $\Phi_\beta$ and $\lambda^\star(\beta)$ are learned on $K-1$ folds, and the estimator is evaluated on the held-out fold. For each fold $k$, the value estimate is

$$\widehat{V}_k = \frac{1}{I_k} \sum_{n \in I_k} \left( Y^{(n)} + \lambda^*(\widehat{\beta})^\top C^{(n)}(\widehat{\beta}) \right). \tag{4}$$

Finally, the overall estimate aggregates across folds,

$$\widehat{V}_{\text{Shaped-PDIS}}^{\pi_e}(\lambda) = \frac{1}{K} \sum_k \widehat{V}_k, \tag{5}$$

with empirical residual variance $\widehat{S}^2 = \frac{1}{K} \sum_k S_k^2$.

## 4.3   RELATION TO DR AND MRDR

It is useful to contrast shaping-based estimators with the doubly robust (DR) and more robust doubly robust (MRDR) estimators. Both DR and MRDR employ the idea of control variates, but they rely on learning an approximate $Q$–function. This is fundamentally more challenging than learning a potential $\Phi_\beta$: the $Q$–function is state–action dependent, must satisfy a Bellman consistency condition, and directly encodes long–horizon returns. By contrast, shaped OPE estimators requires only a scalar potential over states, grounded at terminal states. This yields two key advantages:

**Ease of learning.**    Potentials can be learned by simple regression or by optimizing the variance-reduction objective $J(\beta)$, without requiring value-consistency or bootstrapping. The shaping columns $C^n(\beta)$ are therefore easier to estimate reliably than $Q$-based control variates, especially in sparse–reward regimes.

**Interpretability.**    The potential $\Phi_\beta(s)$ can be interpreted as a measure of "progress" or "risk" associated with a state: high values near successful outcomes and low values near dead–ends. This scalar field provides a transparent explanation of what the variance reduction is "taking out," making shaped estimators easier to audit than DR/MRDR, whose $Q$ estimates are often opaque.

**Tighter concentration via variance reduction.**    PDIS is unbiased but Hoeffding–type bounds are often loose because they scale with worst–case rewards and importance weights. Shaped PDIS reduces the *residual variance* of the PDIS return $Y$ by adding a zero-mean shaping control variate $C^\Phi$, thereby tightening variance-aware bounds. Under $|r_t| \le R, |\Phi| \le B, |W_t| \le w$, Bernstein gives, for any $\delta \in (0, 1)$,

$$\left| \hat{V}_{\text{Shaped-PDIS}}^{\pi_e} - V^{\pi_e} \right| \le \sqrt{\frac{2\,\sigma^2(\lambda)}{N} \log \frac{2}{\delta}} + \frac{2\,L(\lambda)}{3N} \log \frac{2}{\delta}, \quad \sigma^2(\lambda) = \text{Var}[Y + \lambda C^\Phi],$$

where $L(\lambda)$ bounds $|Y + \lambda C^\Phi|$. At the variance–optimal $\lambda^\star$, $\sigma^2(\lambda^\star) \le \text{Var}[Y]$ (strict if $\text{Cov}[Y, C^\Phi] \ne 0$). An empirical Bernstein variant yields

$$\left| \hat{V}_{\text{Shaped-PDIS}}^{\pi_e} - V^{\pi_e} \right| \le \sqrt{\frac{2\,S_N^2(\lambda)}{N} \log \frac{3}{\delta}} + \frac{3\,L(\lambda)}{N} \log \frac{3}{\delta},$$

with $S_N^2(\lambda)$ the sample variance of $\{Y^n + \lambda C^{\Phi,n}\}_{n=1}^N$. This bound adapts to the observed variance, substantially reduced by shaped estimators in sparse-reward domains to yield tighter uncertainty quantification. Analogous bounds for shaped DR/MRDR appear in Appendix E.1.

## 5 EXPERIMENTS

Our experiments are designed to stress-test OPE estimators across domains where standard methods are known to fail: long-horizons, sparse rewards, noisy reward signals, and real-world clinical trajectories. We begin by evaluating on a tabular Chain MDP environment, followed by an ICU-Sepsis benchmark and a medical simulator for Cancer.

**Baselines.** For all domains, we compare the performance of our approach against PDIS, DR and MRDR baselines. For DR, we use a fitted Q evaluation (FQE) critic with sufficient iterations to yield accurate value estimates. MRDR trains a critic using an importance-weighted regression objective to minimize estimator variance within the Q-function class. Finally, each shaping-based estimator employs reward-shaping control variates, where a potential function $\Phi_\beta$ is parameterized and learned from logged data by maximizing the explained variance between PDIS/DR/MRDR returns and shaped control-variate columns respectively. This ensures the resulting control variate is unbiased and directly optimized for variance reduction.

**Metrics.** Throughout this section, we report the mean-squared error (MSE), bias, variance and effective sample size (ESS) to assess the quality of our shaped estimators. Detailed descriptions of these metrics can be found in Appendix F.1. Additional experimental setup details and hyperparameters for each domain can be found in Appendix G.

### 5.1 EVALUATION ON TABULAR CHAIN MDP

**Environment Structure.** We consider a finite-horizon chain MDP with states $\{0, \ldots, S-1\}$, starting at $s_0$. There are two absorbing terminals: success at $s_{20}$ and dead-end at $s_{10}$. Episodes end upon reaching a terminal or after $T = 18$ steps. At non-terminal states, the agent chooses *left* ($a = 0$) or *right* ($a = 1$). *Right* moves forward with $p = 0.9$, else stays; *left* moves backward with $p = 0.2$, else stays. Success yields $+1$, dead-end $-1$, and all other transitions $0$.

**Policies.** The behavior policy $\pi_b$ is conservative, choosing $a = 0$ with $p = 0.7$ and $a = 1$ with $p = 0.3$. The evaluation policy $\pi_e$ is aggressive, with probabilities reversed. This induces distribution shift: $\pi_b$ often stalls or reaches the dead-end, while $\pi_e$ more frequently reaches success.

### 5.2 EVALUATION ON CANCER SIMULATOR

**Environment Structure.** We use the cancer simulator of Ribba et al. (2012), which models tumor progression under chemotherapy via a small system of differential equations. It tracks proliferating and quiescent tumor cells, with treatment effects delayed through the quiescent compartment, and drug concentration in the body. For RL, we discretize time into monthly decision steps. The state space has 4 features (cell counts and drug concentration). Each month a clinician may administer treatment or not. The per-step reward is the change in tumor diameter: $r_t = -(MTD_{t+1} - MTD_t)$, with positive values indicating improvement.

**Policies.** The evaluation policy $\pi_e$ treats patients monthly for 10 months, then stops. The behavior policy $\pi_b$ is an $\epsilon$-greedy variant: with probability $1 - \epsilon$ it follows $\pi_e$, otherwise it takes the opposite action, with $\epsilon \in \{0.1, 0.3, 0.5\}$.

### 5.3 EVALUATION ON ICU-SEPSIS BENCHMARK

**Environment Structure.** We evaluate on the real-world ICU-Sepsis benchmark (Choudhary et al., 2024) derived from MIMIC-III. Patient trajectories are segmented into 4-hour windows over horizons up to 72 hours using the benchmark's standardized pre-processing. Each state $s_t$ has 47 features, including vital signs (e.g., heart rate, blood pressure), labs (e.g., lactate, creatinine), demographics, and derived scores (e.g., SOFA), all normalized and imputed per the benchmark. Actions $a_t$ are intravenous fluids and vasopressors, with dosages discretized into a $5 \times 5$ grid (25 discrete

actions). Rewards are sparse: $+1$ if discharged alive, $0$ if deceased, and $0$ otherwise. This reward structure reflects the clinical challenge of evaluating policies with delayed outcomes.

**Policies.** We generate $\pi_b$ and $\pi_e$ by training PPO for 1M episodes, taking $\pi_b$ as the actor at episode 250k and $\pi_e$ at episode 1M. Their large optimality gap induces distribution shift. The ground-truth value of $\pi_e$ is estimated via Monte Carlo rollouts in the benchmark's learned environment model.

## 6 RESULTS AND DISCUSSION

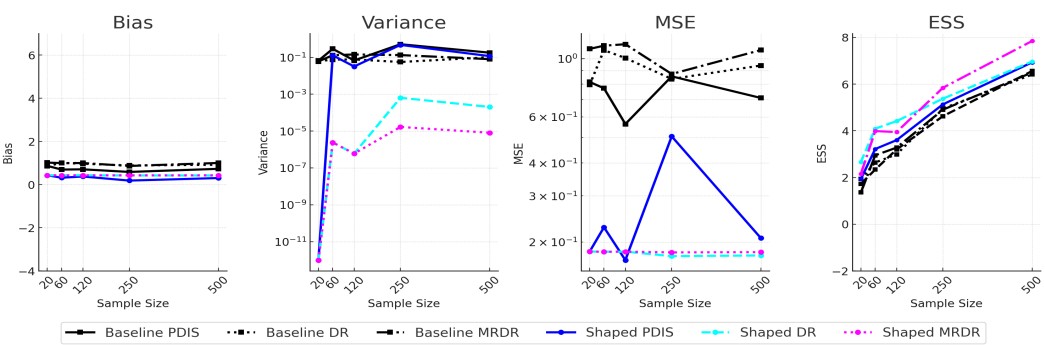

Figure 2: Tabular-Chain. We observe the bias, variance, MSE of the shaped estimators is lower than the bias, variance, MSE of the traditional estimators. Additionally, the ESS of the shaped-estimators has higher mean and lower deviation as compared to the shaped-estimators.

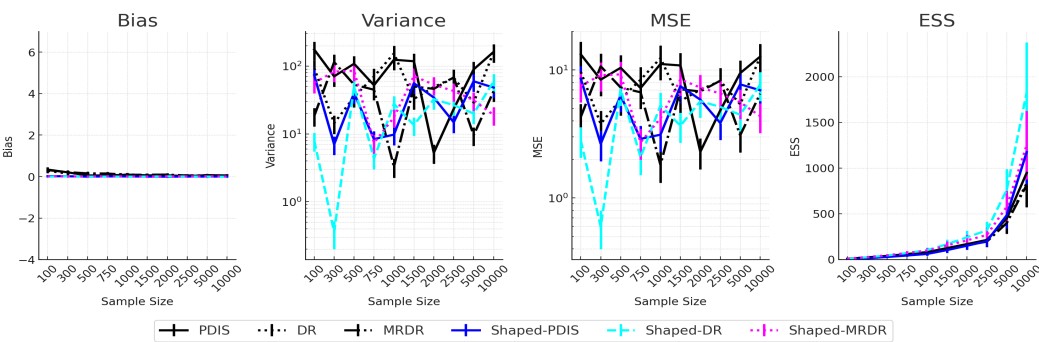

Figure 3: Cancer. We observe the bias, variance, MSE of the shaped estimators is generally lower than the bias, variance, MSE of the traditional estimators. Additionally, the ESS of the shaped-estimators has higher mean and lower deviation as compared to the shaped-estimators.

**Shaped estimators have orders of improvement in Variance, MSE and ESS while being unbiased over traditional estimators.** Figure 2,4 and 3 (Appendix G) compare the performance of the shaped and the traditional variants of the OPE estimators on real-world ICU-Sepsis data. Across all environments, we observe shaped estimators outperform tradtional estimators across all metrics. For ICU-Sepsis, we observe, the variance and MSE of the shaped estimators are 3-4 orders of magnitude better than the traditional estimators, indicating the shaped potential functions $\phi(s)$ successfully provide intermediate signals pertinent to reducing variance. Consequently, we observe the ESS of shaped OPE estimators to be higher than the traditional variants. Consistent to our theoretical findings, we observe the shaped-variants of the OPE estimators remain asymptotically unbiased.

**Shaped estimators are more robust to noise than traditional OPE estimators:** Figure 5 presents the experiment where varying levels of Gaussian noise are added to the final returns of the estimators in ICU-Sepsis. We observe that the shaped variants are consistently more robust than their traditional counterparts, even as the noise level increases. This robustness arises because shaped estimators can

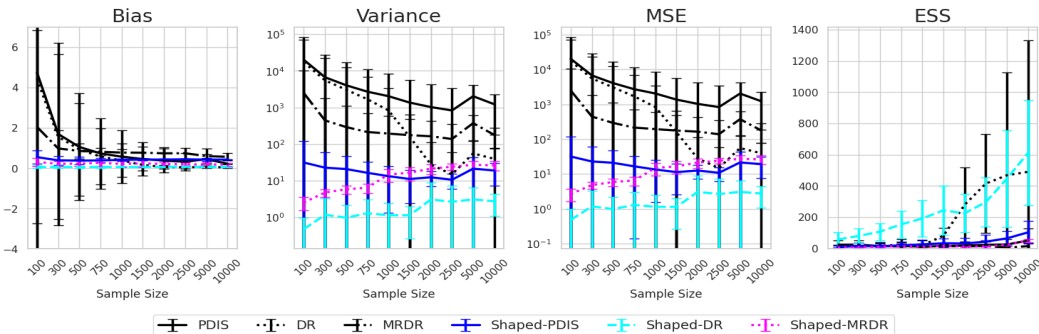

Figure 4: ICU-Sepsis. We observe the variance, MSE of the shaped estimators is lower than the variance, MSE of the traditional estimators, while being unbiased. Additionally, the ESS of the shaped-estimators has higher mean and lower deviation as compared to the shaped-estimators. The experiments were performed for a total of 10 seeds.

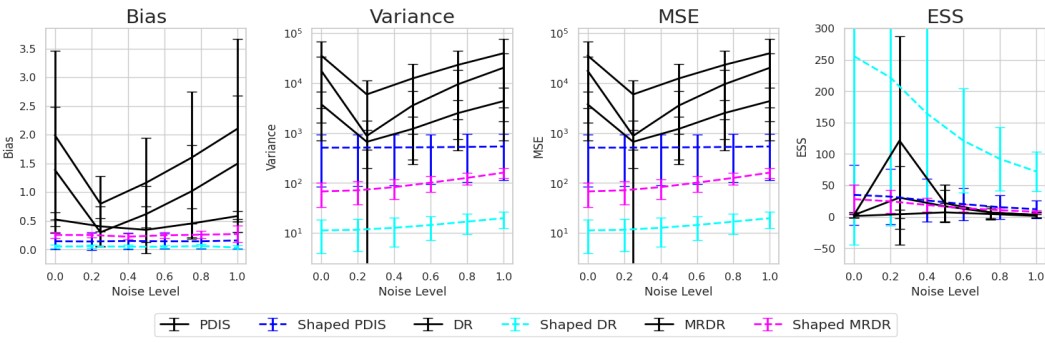

Figure 5: ICU-Sepsis Noise comparison. We observe the shaped estimators are more robust to traditional estimators when the reward signal is contaminated with gaussian noise. The evaluation was conducted over 10k test-examples and for 10 seeds.

leverage intermediate signals from the shaped potential functions, which help mitigate the effect of noise. In contrast, traditional estimators rely on sparse signals, making them more vulnerable to noise, as any corruption in these sparse signals significantly degrades performance. Among the shaped estimators, the DR and MRDR variants demonstrate greater robustness compared to the Shaped-PDIS estimator. This is due to the model-based component in DR and MRDR, which helps counteract the impact of noise, absent in the PDIS estimator.

**Shaped-estimators have less deterioration in performance over unshaped-estimators in sparse reward settings.** All our experiments employ sparse reward environments to explore the performance of OPE. Overall, shaped estimators outperform their non-shaped counterparts. Notably, Shaped-DR and Shaped-MRDR estimators have the least deterioration in performance over Shaped-PDIS in sparse environments. This is directly a consequence of the introduction of explicit reward shaping control variates.

**Shaped estimators outperform Marginalised-IS Estimators in sparse-reward settings.** Marginalised estimators were roped in to counter the curse-of-horizon problem that you typically encounter in the IPS ratios due to very high IPS values. However, it depends on estimating the density ration, which isn't known beforehand and needs to be estimated using DICE algorithms. DICE algorithms attempt to solve a minimax optimization problem which converges only if the loss landscape is convex enough Farnia & Ozdaglar (2020),Zhang et al. (2020b). Most RL environments don't fall in this regime, and even slight errors of density ratio estimations adds large bias into the estimator. We conducted an ablation experiment in Fig 6, wherein we considered DualDICE, GenDICE and GradientDICE as our DICE algorithms and compared them against shaped-PDIS, shaped-DR and

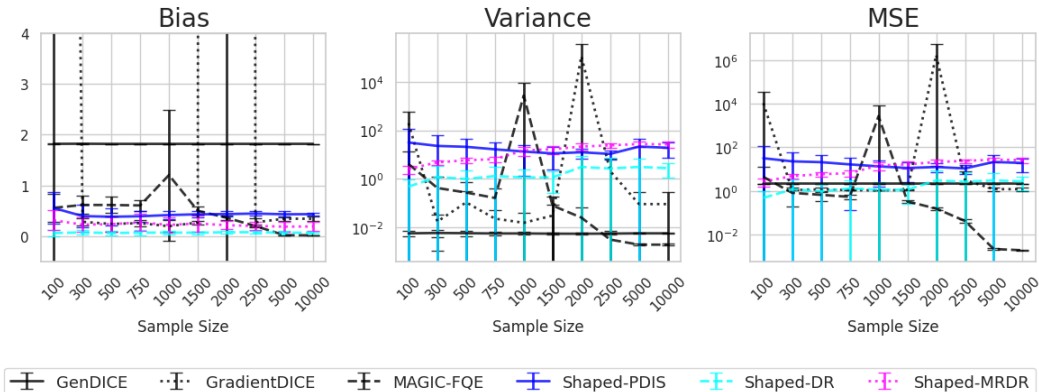

Figure 6: ICU-Sepsis experiment with additional baselines. We consider Magic-FQE as an improved FQE baseline, and use Marginalised-IS estimators generated by GradientDICE and GenDICE. Our shaped estimators are still based out of poor model-based FQE. We observe, while the MIS variants have comparable to lower variance, they are still very high biased as compared to shaped-variants. Shaped estimators perform competitively against Magic-FQE on lower samples inspite of having a bad model based function, indicating the potential shaping functions can rectify bias.

shaped-MRDR. We observe, DualDICE and GenDICE completely collapse due to lack of reward signals (It has been theoretically proven DualDICE algorithm is unstable with large discount factors Zhang et al. (2020a)). GradientDICE algorithm is stabler, has a low variance, but due to inaccuracy in estimating the density ratio, they incorporate a huge bias, and consequently MSE. Even the simplest shaped estimator (Shaped-PDIS) outperforms all the DICE estimators despite incorporating the curse-of-horizon problem, indicating the strength of the potential functions and the algorithm to improve the OPE estimators.

**Shaping potential functions reveal helpful and harmful states and are easier to interpret than DR.** Unlike DR, which learns an *action-conditioned* control variate via $Q(s, a)$, shaping learns a *state-based* control variate from temporal potential differences $\phi(s') - \phi(s)$. In practice, learning $Q(s, a)$ and using it for evaluation using techniques like FQE is a hard problem in itself. $\phi(s)$ is action-agnostic which allows for direct comparison across states, so extremes of $\phi$ immediately rank states by risk in terms of variance/MSE contribution and expose the features that drive it. In our ICU sepsis case, low-$\phi$ states consistently reflected deterioration through low GCS and blood pressure, low $SpO_2$, high $FiO_2$, very high WBC, prolonged PTT, and very high $P_aCO_2$. High-$\phi$ states reflected stability through moderate GCS, normal blood pressure, high $FiO_2$ with moderate $SpO_2$ and $P_aCO_2$, normal WBC, and medium–high bicarbonate ($HCO_3^-$). Tables 1–2 (Appendix H detail these states and their $\phi(s)$. Beyond variance reduction, this clinically legible, state-level signal enables targeted policy improvement, a significant advantage over sparse-reward scenarios, complementing the primary benefit of variance reduction.

## 7 CONCLUSIONS AND FUTURE WORK

This paper introduced reward shaping-based control variates for off-policy evaluation in sparse and noisy reward settings. We derived a family of shaping-based estimators and showed theoretically and empirically that these estimators can both reduce variance and preserve the policy value, while remaining unbiased. Notably, the shaped OPE estimators offer significant advantages in terms of their interpretability as their values directly rank helpful and harmful states and reveal the features that drive them, offering insights that are harder to obtain from doubly robust methods. Future work should explore how shaping-based OPE methods perform in partially observable environments, as well as hybrid shaping-based and marginalized estimators, and prospective studies where reward shaping control variates guide offline policy selection and safe deployment.

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

## A  PRELIMINARIES: EXISTING OPE ESTIMATORS

The goal of our approach is to reduce the variance and MSE of off-policy evaluation in sparse reward settings. Instead of using a value function as a control variate like DR methods, we use a potential-based reward shaping control variate and optimize over the space of shaped rewards. Because of its resemblance to DR, we provide a brief overview of the IS and DR approaches.

**Per-Decision Importance Sampling.**  Importance sampling (IS) estimators reweight observed trajectories to correct for the distribution mismatch between $\pi_b$ and $\pi_e$. The standard trajectory-wise IS estimator is given by

$$\hat{V}_{\text{IS}}^{\pi_e} = \frac{1}{N} \sum_{n=1}^{N} W_{T-1}^{(n)} R_{0:T-1}(\tau^{(n)}); \qquad W_{T-1}^{(n)} = \prod_{k=0}^{T-1} \frac{\pi_e(a_k \mid s_k)}{\pi_b(a_k \mid s_k)}, \qquad (6)$$

is the importance ratio of the trajectory.

This estimator is unbiased but suffers from exponential variance in horizon length. To address this, the *per-decision IS (PDIS)* estimator decomposes the return into stepwise contributions:

$$\widehat{V}_{\text{PDIS}}^{\pi_e} = \frac{1}{N} \sum_{n=1}^{N} \sum_{t=0}^{T-1} \gamma^t W_t^{(n)} r_t^{(n)}, \qquad W_t^{(n)} = \prod_{k=0}^{t} \frac{\pi_e(a_k^{(n)} \mid s_k^{(n)})}{\pi_b(a_k^{(n)} \mid s_k^{(n)})}, \qquad (7)$$

Using partial trajectory overlap, PDIS achieves lower variance than trajectory-wise IS, but still relies on frequent reward observations and performs poorly in sparse-reward settings.

**Doubly Robust Off-Policy Evaluation.**  DR estimators combine DM with IS and have been widely used in regression (Cassel et al., 1976), contextual bandits (Dudík et al., 2011), and RL (e.g. Thomas & Brunskill (2016); Jiang & Li (2016)). In RL, the DR estimate is given by,

$$\hat{V}_{\text{DR}}^{\pi_e}(\beta) = \frac{1}{N} \sum_{n=1}^{N} \sum_{t=0}^{T-1} \gamma^t \Big[ W_t^{(n)} r_t^{(n)} - W_t^{(n)} \hat{Q}^{\pi_e}(s_t^{(n)}, a_t^{(n)}; \beta) + W_{t-1}^{(n)} \hat{V}^{\pi_e}(s_t^{(n)}; \beta) \Big]. \quad (8)$$

The IS part of DR is based on step-IS while the model part relies on $\hat{Q}^{\pi_e}$ and $\hat{V}^{\pi_e}$ model estimates. Importantly, the bias of the DR estimator is a product of both the bias of DM and IS. As a result, DR is unbiased if either IS or DM is unbiased. When the behaviour policy $\pi_b$ is known, Eq. 8 is unbiased. The MRDR estimator (Farajtabar et al., 2018) modifies classic DR by learning the model parameter that minimizes the variance of the DR estimator.

## B  Intuition behind the potential shaping function from the lens of Off-Policy Evaluation

Given an MDP environment, a single task can be represented by the two different kinds of reward functions, one dependent on just states, and one dependent on the $(s, a, s')$ transition. As an example, consider a 1D environment $s_1 - s_2 - s_3$ where $s_3$ is the goal, and the choices of actions are left and right. Now, these reward definitions are equivalent:

Reward function 1 assigns $R(s_3) = 1$ and $R(s_1) = R(s_2) = 0$. In this formulation, a reward is obtained only upon reaching state $s_3$; all other states yield zero reward. Notice that the reward does not depend on actions, the policy gradient algorithm is responsible for learning which actions to take. Reward function 2 instead defines the reward at the transition level: $R(s_2, a_{\text{right}}, s_3) = 1$, $R(s_3, a_{\text{right}}, s_3) = 1$, and 0 for all other transitions. Here, the reward explicitly depends on both the state and the action taken. Although the two reward definitions are equivalent in terms of the task they specify, defining rewards purely as functions of states is often simpler and more convenient.

This idea extends naturally to potential functions $\Phi$. Potential functions can depend only on states, outputting a scalar value that helps determine whether the preceding action was beneficial. Even without taking the action as an explicit input, a state-based potential function can still guide learning effectively because the transition from one state to another implicitly reflects the action taken. Action-based potential functions do exist, for example, Xiao et al. (2019) but keeping $\Phi$ state-based helps preserve task fidelity and avoids unintentionally altering optimality. Moreover, learning action-dependent potential functions typically requires more samples and offers sample efficiency comparable to directly learning a value function, as in Doubly Robust estimator.

## C  Reward Shaping Control Variates

We provide complete proofs of the theory mentioned in Section 4 here.

*Proof of Lemma 1.* For each $t$, by a change of measure using importance weights,

$$\mathbb{E}_{\pi_b}\left[\gamma^t W_t(\gamma\Phi(s_{t+1}) - \Phi(s_t))\right] = \mathbb{E}_{\pi_e}[\gamma^{t+1}\Phi(s_{t+1}) - \gamma^t\Phi(s_t)].$$

Summing over $t = 0, \ldots, T - 1$ gives $\mathbb{E}_{\pi_e}[\gamma^T\Phi(s_T) - \Phi(s_0)]$. Adding the start baseline $\mathbb{E}_{\pi_b}[\Phi(s_0)] = \mathbb{E}_{\pi_e}[\Phi(s_0)]$ and using $\Phi(s_T) = 0$ yields zero. $\square$

$$\mathbb{E}_{\pi_b}\left[\gamma^t W_t\left(\gamma\Phi(s_{t+1}) - \Phi(s_t)\right)\right] = \mathbb{E}_{\pi_b}\left[\gamma^t\left(\prod_{k=0}^{t}\frac{\pi_e(a_k \mid s_k)}{\pi_b(a_k \mid s_k)}\right)(\gamma\Phi(s_{t+1}) - \Phi(s_t))\right] \quad \text{(a)}$$

$$= \frac{1}{N}\sum_{n=1}^{N}\left[\left(\prod_{k=0}^{T}\pi_b(a_k \mid s_k)\right)\left(\gamma^t\left(\prod_{k=0}^{T}\frac{\pi_e(a_k \mid s_k)}{\pi_b(a_k \mid s_k)}\right)(\gamma\Phi(s_{t+1}) - \Phi(s_t))\right)\right] \quad \text{(b)}$$

$$= \frac{1}{N}\sum_{n=1}^{N}\left[\gamma^t\left(\prod_{k=0}^{T}\pi_e(a_k \mid s_k)\right)(\gamma\Phi(s_{t+1}) - \Phi(s_t))\right] \quad \text{(c)}$$

$$= \mathbb{E}_{\pi_e}\left[\gamma^t\left(\gamma\Phi(s_{t+1}) - \Phi(s_t)\right)\right] \quad \text{(d)}$$

**Explanation of steps:**

(a) Expanding the importance sampling ratio $W_t$.

(b) Expanding the definition of expectation under the behavior policy.

(c) $\prod_{t=0}^{T}\pi_b(a_t \mid s_t)$ is the probability of the trajectory. This probability cancels with the denominator of $W_t$.

(d) Pulling $\prod_{t=0}^{T}\pi_e(a_t \mid s_t)$ outside the summation and re-writing it as expectation under the evaluation policy. This completes the lemma.

*Proof of Theorem 3.* Since $\mathbb{E}[C^\Phi] = 0$, we have

$$\mathbb{E}\big[\widehat{V}_{\text{Shaped}-\text{PDIS}}(\lambda)\big] = \frac{1}{N}\sum_{i=1}^{N}\mathbb{E}[Y_i + \lambda C_i^\Phi] = \mathbb{E}[Y],$$

so the estimator remains unbiased. By independence across episodes,

$$\text{Var}\big[\widehat{V}_{\text{Shaped}-\text{PDIS}}(\lambda)\big] = \frac{1}{N}\text{Var}\big[Y + \lambda C^\Phi\big].$$

Expanding,

$$\text{Var}[Y + \lambda C^\Phi] = \text{Var}[Y] + \lambda^2\text{Var}[C^\Phi] + 2\lambda\text{Cov}[Y, C^\Phi],$$

which yields Eq. 2. This is a convex quadratic in $\lambda$, minimized at

$$\lambda^\star = -\frac{\text{Cov}[Y, C^\Phi]}{\text{Var}[C^\Phi]}.$$

Substituting this value back in and completing the square gives

$$\text{Var}\big[\widehat{V}_{\text{Shaped}-\text{PDIS}}(\lambda)\big] = \frac{1}{N}\left(\text{Var}[Y] - \frac{\text{Cov}[Y, C^\Phi]^2}{\text{Var}[C^\Phi]} + \text{Var}[C^\Phi]\Big(\lambda + \frac{\text{Cov}[Y, C^\Phi]}{\text{Var}[C^\Phi]}\Big)^2\right).$$

At $\lambda = \lambda^\star$, the squared term vanishes. Since $\text{Cov}[Y, C^\Phi]^2/\text{Var}[C^\Phi] \geq 0$, the minimized variance is no larger than $\text{Var}[Y]/N$, with equality if and only if $\text{Cov}[Y, C^\Phi] = 0$. $\qquad\square$

## D    A FAMILY OF SHAPED ESTIMATORS AND THEIR PROPERTIES

In addition to the standard Shaped PDIS estimator we present in the main paper, the idea of reward shaping control variates can be integrated into existing OPE estimators with similar performance guarantees in sparse reward settings. Here, we show in particular how RSCV can be integrated into WIS, DR and MRDR estimators.

**Lemma 2** (Per-step zero-mean for WIS control variate). *If $\mathbb{E}_{s_0 \sim d_0^e}[\Phi(s_0)] = 0$ and $\Phi(s_T) = 0$ on terminals, then for every $t$, $\mathbb{E}_{\pi_e}[c_t^\Phi] = 0$ and hence $\mathbb{E}_{\pi_b}[\rho_{0:t}c_t^\Phi] = 0$.*

*Proof.* Telescoping gives $\sum_{t=0}^{T-1} c_t^\Phi = \gamma^T\Phi(s_T) - \Phi(s_0)$. Taking $\mathbb{E}_{\pi_e}$ and using $\Phi(s_T) = 0$ yields $\sum_t \mathbb{E}_{\pi_e}[c_t^\Phi] = -\mathbb{E}_{d_0^e}[\Phi(s_0)] = 0$. Since $\sum_t$ is zero and each $c_t^\Phi$ is integrable, a sufficient condition (enforced by centering at $t = 0$) is $\mathbb{E}_{\pi_e}[c_t^\Phi] = 0$ for all $t$. The equality $\mathbb{E}_{\pi_b}[\rho_{0:t}c_t^\Phi] = \mathbb{E}_{\pi_e}[c_t^\Phi]$ follows by change of measure. $\qquad\square$

**Lemma 3** (Per-step zero-mean for DR/MRDR control variate). *If $\mathbb{E}_{s_0 \sim d_0}[\Phi(s_0)] = 0$ for the common initial distribution of $\pi_b$ and $\pi_e$, and $\Phi(s_T) = 0$, then for every $t$, $\mathbb{E}_{\pi_b}[\rho_{0:t}c_t^\Phi] = \mathbb{E}_{\pi_e}[c_t^\Phi] = 0$.*

*Proof.* For any integrable $f_t$, $\mathbb{E}_{\pi_b}[\rho_{0:t}f_t] = \mathbb{E}_{\pi_e}[f_t]$. With $f_t = c_t^\Phi$ and telescoping as above, $\sum_t \mathbb{E}_{\pi_e}[c_t^\Phi] = -\mathbb{E}_{s_0 \sim d_0}[\Phi(s_0)] = 0$, and centering at $t = 0$ ensures $\mathbb{E}_{\pi_e}[c_t^\Phi] = 0$ for all $t$. $\qquad\square$

**Definition 4** (**Shaped PDWIS estimator**). *Given $N$ i.i.d. trajectories, the shaped per-decision WIS estimator augments PDWIS with a reward shaping control variate $C^\Phi$:*

$$\widehat{V}_{\text{Shaped-PDWIS}}(\lambda) = \sum_{t=0}^{T-1}\sum_{i=1}^{N}\tilde{w}_{i,t}\Big(\gamma^t r_{i,t} + \lambda c_{i,t}^\Phi\Big), \quad \tilde{w}_{i,t} = \frac{\rho_{i,0:t}}{\sum_{j=1}^{N}\rho_{j,0:t}},$$

*where $\lambda \in \mathbb{R}$ is a coefficient controlling the weight of the zero-mean control variate applied.*

**Theorem 4** (**Bias of Shaped PDWIS**). *Under Assumptions and Lemma 2, $\widehat{V}_{\text{Shaped-PDWIS}}(\lambda)$ is asymptotically unbiased:*

$$\lim_{N \to \infty}\mathbb{E}[\widehat{V}_{\text{Shaped-PDWIS}}(\lambda)] = V^{\pi_e}.$$

*It retains the same $O(1/N)$ finite-sample bias as per-step WIS.*

*Proof.* Write $X_{i,t} = \gamma^t r_{i,t} + \lambda c_{i,t}^\Phi$, $W_{i,t} = \rho_{i,0:t}$. Each per-step term is a ratio estimator

$$\hat{\mu}_t = \frac{\sum_i W_{i,t} X_{i,t}}{\sum_i W_{i,t}}.$$

By the Law of Large Numbers and Slutsky's theorem,

$$\hat{\mu}_t \xrightarrow{p} \frac{\mathbb{E}_{\pi_b}[W_{0:t} X_t]}{\mathbb{E}_{\pi_b}[W_{0:t}]} = \mathbb{E}_{\pi_e}[X_t].$$

Summing over $t$ gives $\sum_t \mathbb{E}_{\pi_e}[X_t]$. By Lemma 2, $\mathbb{E}_{\pi_e}[c_t^\Phi] = 0$, hence the limit is $V^{\pi_e}$. Finite-sample bias of ratio estimators is $O(1/N)$ and unaffected by adding a zero-mean control variate. $\square$

**Theorem 5 (Variance of Shaped PDWIS).** *Conditioning on weights, define*

$$A_t = \gamma^t r_t, \quad B_t = c_t^\Phi, \quad \alpha_t = \sum_{i=1}^N \tilde{w}_{i,t}^2.$$

*Then*

$$\mathrm{Var}[\widehat{V}_{\mathrm{Shaped-PDWIS}}(\lambda) \mid \{\tilde{w}\}] = \sum_{t=0}^{T-1} \alpha_t \big(\mathrm{Var}_{w,t}[A_t] + 2\lambda \mathrm{Cov}_{w,t}[A_t, B_t] + \lambda^2 \mathrm{Var}_{w,t}[B_t]\big).$$

*Proof.* Define

$$\mathbf{a} = \sum_t \alpha_t \mathrm{Var}_{w,t}[A_t], \quad \mathbf{b} = \sum_t \alpha_t \mathrm{Cov}_{w,t}[A_t, B_t], \quad \mathbf{c} = \sum_t \alpha_t \mathrm{Var}_{w,t}[B_t].$$

Then the conditional variance is

$$\mathbf{a} + 2\lambda \mathbf{b} + \lambda^2 \mathbf{c}.$$

Add and subtract $\mathbf{b}^2/\mathbf{c}$:

$$= \left(\mathbf{a} - \frac{\mathbf{b}^2}{\mathbf{c}}\right) + \mathbf{c}\left(\lambda + \frac{\mathbf{b}}{\mathbf{c}}\right)^2.$$

Thus minimized at

$$\lambda^\star = -\frac{\mathbf{b}}{\mathbf{c}}, \quad \min_\lambda = \mathbf{a} - \frac{\mathbf{b}^2}{\mathbf{c}}.$$

This is never larger than the baseline $\mathbf{a}$. $\square$

**Definition 5 (Shaped DR estimator).** *Given $N$ i.i.d. trajectories, the shaped DR estimator augments DR with a reward shaping control variate $C^\Phi$:*

$$Z_t^{\mathrm{DR}} = \rho_{0:t}\big(r_t + \gamma \widehat{V}(s_{t+1}) - \widehat{Q}(s_t, a_t)\big), \quad C_t^\Phi = \rho_{0:t} c_t^\Phi,$$

$$\widehat{V}_{\mathrm{Shaped\text{-}DR}}(\lambda) = \frac{1}{N}\sum_{i=1}^N \left[\widehat{V}(s_0^i) + \sum_{t=0}^{T-1}(Z_{i,t}^{\mathrm{DR}} + \lambda C_{i,t}^\Phi)\right],$$

*where $\lambda \in \mathbb{R}$ is a coefficient controlling the weight of the zero-mean control variate applied.*

**Theorem 6 (Bias of Shaped DR).** *Under Assumptions, Lemma 3, and standard DR conditions, $\widehat{V}_{\mathrm{Shaped\text{-}DR}}(\lambda)$ is unbiased:*

$$\mathbb{E}[\widehat{V}_{\mathrm{Shaped\text{-}DR}}(\lambda)] = V^{\pi_e}.$$

*Proof.* By DR unbiasedness, $\mathbb{E}[\widehat{V}(s_0) + \sum_t Z_t^{\mathrm{DR}}] = V^{\pi_e}$. By Lemma 3, $\mathbb{E}[C_t^\Phi] = 0$, hence $\mathbb{E}[\sum_t C_t^\Phi] = 0$. Therefore, $E[\widehat{V}_{\mathrm{Shaped-DR}(\lambda)}] = V^{\pi_e}$. $\square$

**Theorem 7 (Variance of Shaped DR).** *Let $Y = \widehat{V}(s_0) + \sum_t Z_t^{\mathrm{DR}}$, $C = \sum_t C_t^\Phi$. Then*

$$\mathrm{Var}[\widehat{V}_{\mathrm{Shaped\text{-}DR}}(\lambda)] = \frac{1}{N}\Big(\mathrm{Var}[Y] + 2\lambda \mathrm{Cov}[Y, C] + \lambda^2 \mathrm{Var}[C]\Big).$$

*Proof.* For one trajectory,
$$\mathrm{Var}[Y + \lambda C] = \mathrm{Var}[Y] + 2\lambda\mathrm{Cov}[Y, C] + \lambda^2\mathrm{Var}[C].$$
Add and subtract $\mathrm{Cov}[Y, C]^2/\mathrm{Var}[C]$:
$$= \left(\mathrm{Var}[Y] - \tfrac{\mathrm{Cov}[Y,C]^2}{\mathrm{Var}[C]}\right) + \mathrm{Var}[C]\left(\lambda + \tfrac{\mathrm{Cov}[Y,C]}{\mathrm{Var}[C]}\right)^2.$$
This is minimized at
$$\lambda^\star = -\frac{\mathrm{Cov}[Y, C]}{\mathrm{Var}[C]}, \quad \min_\lambda = \mathrm{Var}[Y] - \frac{\mathrm{Cov}[Y, C]^2}{\mathrm{Var}[C]}.$$
Dividing by $N$ gives the estimator variance. $\qquad\square$

**Definition 6** (**Shaped MRDR estimator**). *Given $N$ i.i.d. trajectories, the shaped MRDR estimator augments MRDR with a reward shaping control variate $C^\Phi$:*
$$Z_t^{\mathrm{MRDR}} = \rho_{0:t}\big(r_t + \gamma\widehat{V}_{\mathrm{MRDR}}(s_{t+1}) - \widehat{Q}_{\mathrm{MRDR}}(s_t, a_t)\big), \quad C_t^\Phi = \rho_{0:t}\, c_t^\Phi,$$
$$\widehat{V}_{\mathrm{Shaped\text{-}MRDR}}(\lambda) = \tfrac{1}{N}\sum_{i=1}^N \left[\widehat{V}_{\mathrm{MRDR}}(s_0^i) + \sum_{t=0}^{T-1}(Z_{i,t}^{\mathrm{MRDR}} + \lambda\, C_{i,t}^\Phi)\right],$$
*where $\lambda \in \mathbb{R}$ is a coefficient controlling the weight of the zero-mean control variate applied.*

**Theorem 8** (**Bias of Shaped-MRDR**). *Under Assumptions, Lemma 3, and MRDR unbiasedness, $\widehat{V}_{\mathrm{Shaped\text{-}MRDR}}(\lambda)$ is unbiased:*
$$\mathbb{E}[\widehat{V}_{\mathrm{Shaped\text{-}MRDR}}(\lambda)] = V^{\pi_e}.$$

*Proof.* MRDR is an instance of DR with variance-minimizing models. Hence by the same reasoning as Theorem 6, expectation equals $V^{\pi_e}$. $\qquad\square$

**Theorem 9** (**Variance of Shaped–MRDR**). *Let $Y = \widehat{V}_{\mathrm{MRDR}}(s_0) + \sum_t Z_t^{\mathrm{MRDR}}$, $C = \sum_t C_t^\Phi$. Then*
$$\mathrm{Var}[\widehat{V}_{\mathrm{Shaped\text{-}MRDR}}(\lambda)] = \tfrac{1}{N}\Big(\mathrm{Var}[Y] + 2\lambda\mathrm{Cov}[Y, C] + \lambda^2\mathrm{Var}[C]\Big).$$

*Proof.* Identical to Theorem 7. Completing the square:
$$\mathrm{Var}[Y + \lambda C] = \left(\mathrm{Var}[Y] - \tfrac{\mathrm{Cov}[Y,C]^2}{\mathrm{Var}[C]}\right) + \mathrm{Var}[C]\left(\lambda + \tfrac{\mathrm{Cov}[Y,C]}{\mathrm{Var}[C]}\right)^2.$$
Thus
$$\lambda^\star = -\frac{\mathrm{Cov}[Y, C]}{\mathrm{Var}[C]}, \quad \min_\lambda = \mathrm{Var}[Y] - \frac{\mathrm{Cov}[Y, C]^2}{\mathrm{Var}[C]}.$$
$$\square$$

## E    REWARD SHAPING CONTROL VARIATES FOR VARIANCE REDUCTION

**Details for Learning Control Variates for Variance Guarantees.** Recall from Section 4, when optimizing control variates $C^\Phi$ for variance reduction, we require computing covariances $\Sigma_{CC}$ and $\Sigma_{CY}$ to optimize the explained variance objective in Eqn. 3. When both $N$ and $T$ are large, explicitly inverting $\Sigma_{CC}$ is infeasible. Instead, we solve the linear system
$$\Sigma_{CC}(\beta)\, v(\beta) = \Sigma_{CY}(\beta), \tag{9}$$
for $v(\beta)$, using the *conjugate gradient (CG)* method, which only requires matrix-vector products. In this formulation,
$$J(\beta) = \Sigma_{CY}(\beta)^\top v(\beta), \qquad \lambda^\star(\beta) = -v(\beta). \tag{10}$$
Because $\Sigma_{CC}(\beta)$ is symmetric positive-definite (a covariance plus ridge), conjugate gradients converge rapidly in practice. The matrix–vector product is implemented implicitly as
$$x \mapsto \Sigma_{CC}(\beta)\, x = \alpha x + \tfrac{1}{N}C(\beta)^\top\big(C(\beta)x\big), \tag{11}$$
which requires only two passes through the control variates matrix $C(\beta)$. This allows minibatching over trajectories, avoiding the need to materialize $\Sigma_{CC}$. Summarily, the efficiency of our approach relies critically on the fact that the estimator is *linear in $\lambda$*: this structure reduces variance minimization to solving a symmetric positive-definite linear system, which can be handled scalably with conjugate gradients.

### E.1 Confidence Bounds of Shaped Estimators

In Section 4, we provided theoretical bounds on the performance of Shaped-PDIS, demonstrating that these performance bounds were tighter than those of standard PDIS. However, since standard OPE estimators often have vacuous high-probability bounds, in this section, we provide tighter bounds for Shaped-DR and Shaped-MRDR estimators in comparison to their non-shaped counterparts.

**Confidence bounds of Shaped DR Estimator.** Assume boundedness (for some $R, B, w > 0$) ensuring a uniform envelope

$$|Y_{\text{DR}} + \lambda C| \leq L_{\text{DR}}(\lambda), \qquad \forall \lambda \in \mathbb{R}. \tag{12}$$

Let

$$\sigma_{\text{DR}}^2(\lambda) := \text{Var}\big[Y_{\text{DR}} + \lambda C\big]. \tag{13}$$

Then, for any $\delta \in (0, 1)$, Bernstein's inequality yields

$$\big|\widehat{V}_{\text{Shaped-DR}}(\lambda) - V^{\pi_e}\big| \leq \sqrt{\frac{2\,\sigma_{\text{DR}}^2(\lambda)}{N}\log\frac{2}{\delta}} + \frac{2\,L_{\text{DR}}(\lambda)}{3N}\log\frac{2}{\delta}. \tag{14}$$

At $\lambda = \lambda_{\text{DR}}^\star$, we have $\sigma_{\text{DR}}^2(\lambda_{\text{DR}}^\star) \leq \text{Var}[Y_{\text{DR}}]$, so the Bernstein CI for Shaped–DR is (weakly) tighter than that for DR at the same confidence.

Let $S_N^2(\lambda)$ denote the sample variance of $\{Y_{\text{DR},i} + \lambda C_i\}_{i=1}^N$. The empirical Bernstein bound gives

$$\big|\widehat{V}_{\text{Shaped-DR}}(\lambda) - V^{\pi_e}\big| \leq \sqrt{\frac{2\,S_N^2(\lambda)}{N}\log\frac{3}{\delta}} + \frac{3\,L_{\text{DR}}(\lambda)}{N}\log\frac{3}{\delta}. \tag{15}$$

At $\lambda_{\text{DR}}^\star$, $S_N^2(\lambda_{\text{DR}}^\star)$ concentrates below the DR sample variance, so the empirical intervals are likewise tighter.

**Confidence bounds of Shaped MRDR Estimator.** Assume a uniform envelope

$$|Y_{\text{MRDR}} + \lambda C| \leq L_{\text{MRDR}}(\lambda), \qquad \forall \lambda \in \mathbb{R}. \tag{16}$$

Let

$$\sigma_{\text{MRDR}}^2(\lambda) := \text{Var}\big[Y_{\text{MRDR}} + \lambda C\big]. \tag{17}$$

Then, for any $\delta \in (0, 1)$,

$$\big|\widehat{V}_{\text{Shaped-MRDR}}(\lambda) - V^{\pi_e}\big| \leq \sqrt{\frac{2\,\sigma_{\text{MRDR}}^2(\lambda)}{N}\log\frac{2}{\delta}} + \frac{2\,L_{\text{MRDR}}(\lambda)}{3N}\log\frac{2}{\delta}. \tag{18}$$

At $\lambda_{\text{MRDR}}^\star$, $\sigma_{\text{MRDR}}^2(\lambda_{\text{MRDR}}^\star) \leq \text{Var}[Y_{\text{MRDR}}]$, so the Bernstein CI for Shaped–MRDR is (weakly) tighter than that for MRDR.

Let $S_N^2(\lambda)$ be the sample variance of $\{Y_{\text{MRDR},i} + \lambda C_i\}_{i=1}^N$. The empirical Bernstein inequality implies

$$\big|\widehat{V}_{\text{Shaped-MRDR}}(\lambda) - V^{\pi_e}\big| \leq \sqrt{\frac{2\,S_N^2(\lambda)}{N}\log\frac{3}{\delta}} + \frac{3\,L_{\text{MRDR}}(\lambda)}{N}\log\frac{3}{\delta}. \tag{19}$$

Evaluated at $\lambda_{\text{MRDR}}^\star$, these intervals are tighter.

## F Experiment Details

### F.1 Evaluation Metrics

Here we describe some of the key evaluation metrics we use to assess the validity of our shaped estimators. These metrics have been widely used across works that focus on OPE e.g. (Dudík et al., 2011; Farajtabar et al., 2018; Thomas & Brunskill, 2016).

**True policy value.** The value of the evaluation policy $\pi_e$ is

$$V(\pi_e) = \mathbb{E}_{\tau \sim \pi_e} \left[ \sum_{t=0}^{T} \gamma^t r_t \right],$$

where $\tau = (s_0, a_0, r_0, \ldots, s_T)$ denotes a trajectory.

**Bias.** Bias quantifies the systematic deviation of an estimator from the true policy value. Let $\hat{V}(\pi_e)$ denote an estimate of $V(\pi_e)$. The bias is defined as

$$\text{Bias} = \mathbb{E}_{\tau \sim \pi_b} \left[ \hat{V}(\pi_e) \right] - V(\pi_e),$$

where the expectation is taken over trajectories generated by $\pi_b$, and any additional randomness in the estimator. In practice, we approximate bias by computing the mean difference between OPE estimates and the ground-truth policy value over repeated runs.

**Variance.** Variance measures the spread or instability of the estimator around its expected value under $\pi_b$. Formally,

$$\text{Var} = \mathbb{E}_{\tau \sim \pi_b} \left[ \left( \hat{V}(\pi_e) - \mathbb{E}_{\tau \sim \pi_b}[\hat{V}(\pi_e)] \right)^2 \right].$$

In empirical evaluation, variance is approximated across multiple independent runs of the estimator. A low variance indicates consistent estimates across runs, whereas a high variance implies sensitivity to randomness in data or weight magnitudes.

**Effective Sample Size (ESS).** ESS is a diagnostic metric for the stability and reliability of importance sampling–based estimators. It reflects how many "independent and identically distributed" samples remain after reweighting the dataset. For a dataset of $n$ trajectories sampled under $\pi_b$, let

$$\rho_i = \prod_{t=1}^{T} \frac{\pi_e(a_t|s_t)}{\pi_b(a_t|s_t)}, \quad w_i = \frac{\rho_i}{\sum_{j=1}^{n} \rho_j}.$$

Then,

$$\text{ESS} = \frac{1}{\sum_{i=1}^{n} w_i^2}.$$

The ESS ranges between $1$ and $n$. A high ESS indicates that reweighting distributes influence across many trajectories, whereas a low ESS implies that only a few trajectories dominate, leading to high-variance estimates.

# G ADDITIONAL EXPERIMENT DETAILS AND RESULTS

## G.1 TABULAR CHAIN

**Hyperparameters.** We use a discount factor of $\gamma = 0.97$, three-fold cross-fitting for control variates and ridge regularization with $\alpha = 10^{-2}$ for covariance inversions. Each setting is repeated across 10–20 random seeds. We report results in terms of Bias, Variance, ESS and MSE.

**Extended Experiment** We extend the chain MDP with a controllable reward density parameter to vary the degree of sparsity. Rewards remain terminal-only, with $+1$ at the success state $s_{20}$ and $-1$ at the dead-end $s_{10}$, while all other transitions yield 0. To introduce sparsity, we retain each nonzero terminal reward with probability $d_{\text{term}} \in (0, 1]$ and drop it to zero otherwise. Thus, $d_{\text{term}} = 1$ recovers the standard setting where the agent always observes the terminal signal, while smaller values reduce the frequency of observed rewards, making feedback increasingly sparse. As in the base setup, surviving terminal rewards are corrupted with zero-mean Gaussian noise of standard deviation $\sigma = 0.5$. This construction preserves the structure of the task while enabling a controlled sweep over reward sparsity.

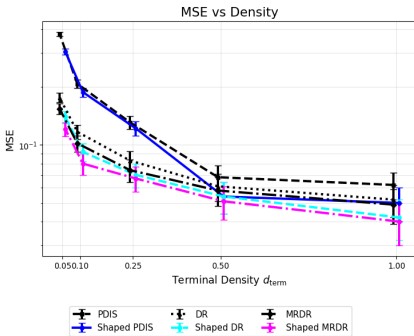

Figure 7: Performance of OPE methods across varying degrees of sparsity of rewards. Shaping-based OPE methods have lower MSE across increasing sparsity. The most extreme setting where a reward is only given at the terminal state is shown in the main body of the paper Figure 2.

.

## G.2 ICU-SEPSIS

**Environment Structure.** We evaluate on the ICU-Sepsis benchmark (Choudhary et al., 2024) derived from the MIMIC-III database, which models the treatment of septic patients. The environment follows the standardized pre-processing provided by the benchmark repository, where patient trajectories are segmented into 4-hour windows over a total horizon of up to 72 hours. Each state $s_t$ consists of approximately 40 features, including vital signs (e.g. heart rate, blood pressure), laboratory values (e.g. lactate, creatinine), demographics and derived scores such as SOFA. These features are normalized and imputed following the pre-processing pipeline in the benchmark. Actions $a_t$ correspond to intravenous fluids and vasopressors that the physician can administer. Following prior work, continuous dosages are discretized into a $5 \times 5$ grid, yielding a 25-dim discrete action space that balances granularity with tractability. The reward signal is sparse and delayed: patients receive a terminal reward of $+1$ if discharged alive and $-1$ if deceased. Intermediate rewards are set to 0. This reward structure reflects the clinical challenge of evaluating policies with delayed outcomes.

**Policies.** We run a PPO algorithm for 1M episodes and select the behavior policy $\pi_b$ as the model parameters of the actor at episode 250k and evaluation policy $\pi_e$ as model parameters at episode 1M. The ground-truth value of $\pi_e$ is approximated by Monte Carlo rollouts in the learned environment model released with the benchmark. The authors in Choudhary et al. (2024) constructed an ICU-Sepsis environment from the real world MIMIC data, wherein the authors used the empirical clinician actions to guide the model training process. We are using the environment directly, upon which we run PPO to get our policies. For further details on the environment, we refer the readers to the original paper Choudhary et al. (2024).

**Hyperparameters.** In order to obtain the policies, we train a PPO for 1M episodes with 1k max-steps using a learning rate of $5e^{-3}$. The other hyperparameters are as follows: $\gamma = 1.0, \lambda_{gae} = 0.4$, update-epochs:6, norm-adv: true, clip-coef: 0.5, clip-vloss: false, ent-coef: 0.005, vf-coef: 0.3,maxgrad-norm: 0.4 and target-kl: 0.001. In our evaluation experiments, we use a discount factor of $\gamma = 1.00$, horizon $T \approx 18$ steps (4h intervals over 72hrs), use dataset size of 10,000 patient trajectories to learn the shaping control variate. RSCV potentials $\Phi_\beta$ are parameterized as two-layer MLPs (128-128, tanh) and trained for 10k Adam steps with ridge regularization $\alpha = 10^{-4}$. FQE critics are trained for 200 epochs with batch size 256 and learning rate $3 \times 10^{-4}$. We evaluate each experiment over a set of [100,300,500,750,1k,1.5k,2k,5k,10k] episodes and repeat it over 10 random seeds, reporting MSE relative to the true values of $\pi_e$, empirical variance and ESS.

## G.3 CANCER SIMULATOR

**Hyperparameters.** For all cancer simulator experiments, we fix the horizon to 30 months with a discount factor of $\gamma = 0.99$. Datasets contain $N \in \{100, 300, 1000, 2000, 5000, 10000\}$ trajectories. Measurement noise on tumor size is modeled as Gaussian with standard deviation $\sigma \in \{0, 1, 2\}$ mm. For RSCV, the potential function $\Phi_\beta$ is parameterized as a one-hidden-layer neural network

with 32 tanh units, trained for 2000 steps using Adam on the explained-variance objective, with ridge regularization $\alpha = 10^{-2}$ and 3-fold cross-fitting for coefficient estimation. The DR critic is trained with FQE (50 iterations for tabular), while MRDR reweights updates using absolute importance ratios for 20 epochs. We evaluate all estimators over 10 random seeds.

# H DETAILED QUALITATIVE INTERPRETATIONS OF SHAPED-POTENTIAL FUNCTIONS IN ICU-SEPSIS

Table 1: Feature descriptions of the top 10 states with the lowest values of shaped-PDIS potential function in ICU-Sepsis. Red indicated bad attributes. We observe these states have Low GCS, Low BP, Low SpO2 levels, High FiO2, Very high White Blood Cell (WBC) count, High Partial Thromboplastin time and Very high partial pressure of arterial carbon dioxide (PaCO2).

| State Feature | $\phi(s)=-1.39$ | $\phi(s)=-1.36$ | $\phi(s)=-1.18$ | $\phi(s)=-1.01$ | $\phi(s)=-0.98$ | $\phi(s)=-0.97$ | $\phi(s)=-0.95$ | $\phi(s)=-0.92$ | $\phi(s)=-0.84$ | $\phi(s)=-0.84$ |
|---|---|---|---|---|---|---|---|---|---|---|
| age | Low | Medium | High | High | Low | Medium | Low | Low | Medium | Low |
| admissionweight | Low | Medium | Medium | Medium | High | Medium | Medium | Low | Medium | High |
| gcs | Low | Low | Low | Low | High | Low | Medium | Low | Low | Medium |
| hr | High | High | Low | Medium | Low | High | High | Medium | Low | High |
| sysbp | Low | Low | Low | Low | Low | Medium | Low | High | High | High |
| meanbp | Low | Low | Low | Low | Low | Medium | Low | High | High | High |
| diabp | Low | Low | Low | Low | Low | Medium | Low | High | High | High |
| rr | High | High | Low | Medium | Low | Medium | High | Medium | Low | Medium |
| temp-c | High | High | Low | Medium | Low | Low | High | Low | Medium | Medium |
| fio2both | High | High | High | High | High | High | High | Low | Low | Medium |
| potassium | Low | Medium | High | Medium | Medium | High | Low | High | Low | Medium |
| sodium | Low | Low | Low | High | Medium | Medium | High | Low | Medium | Medium |
| chloride | Low | Medium | Medium | Low | Low | Medium | High | Medium | Medium | Low |
| glucose | Medium | High | Medium | High | High | High | Low | Low | Low | High |
| magnesium | Medium | Medium | High | High | High | Medium | Medium | Medium | Low | Medium |
| calcium | Medium | Low | Low | High | High | Medium | High | Low | Medium | High |
| hb | Low | High | Medium | Medium | High | High | Medium | Low | Medium | High |
| wbc_count | High | High | High | High | High | Medium | High | Low | Low | Medium |
| platelets_count | High | High | High | High | High | High | Low | Low | Medium | High |
| ptt | High | High | High | High | Low | High | Medium | High | Low | Medium |
| pt | High | High | High | High | High | High | Medium | High | Low | Low |
| arterial_ph | High | Low | Low | Medium | Low | Low | High | Low | Medium | Low |
| pao2 | Low | Low | Low | Low | Medium | Medium | Low | High | High | Medium |
| paco2 | High | High | High | High | High | Low | Low | High | High | High |
| arterial-be | High | Low | Low | High | Medium | Low | Medium | Low | Medium | Low |
| hco3 | High | Low | Low | High | High | Low | Low | Low | High | High |
| arterial_lactate | Medium | High | Low | Medium | High | High | Medium | Low | Low | Medium |
| sofa | High | High | High | High | Low | Medium | High | Medium | High | Medium |
| sirs | High | High | High | High | Low | Medium | High | High | Low | Medium |
| shock-index | High | High | Low | High | Low | High | Medium | Low | Low | Medium |
| pao2-fio2 | High | Low | Low | Low | Low | Medium | Medium | High | High | Medium |
| cumulated-balance_tev | Medium | High | Medium | Low | Low | Medium | Medium | Low | Low | Low |
| spo2 | Low | Low | Medium | Low | Low | Medium | Low | High | Medium | Low |
| bun | Medium | Medium | High | High | Medium | Medium | High | High | Low | Low |
| creatinine | Low | High | High | Medium | Medium | Medium | High | High | Low | Low |
| sgot | High | High | Low | Medium | High | Low | High | Low | Medium | Low |
| sgpt | High | High | Low | Medium | High | Low | High | Low | Medium | Low |
| total-bili | High | Medium | Low | Low | Medium | Medium | High | Low | Medium | Low |
| inr | High | High | High | High | Medium | High | Medium | High | Low | Low |
| input_total_tev | Low | High | Low | Low | Low | Low | Medium | Low | Medium | Low |
| input-4hourly_tev | Low | High | Medium | Medium | Low | Low | Low | Low | Low | Low |
| output-total | Low | Low | Low | High | High | Low | Low | Low | High | Low |
| output-4hourly | Low | Low | Low | Medium | High | Low | Low | Low | Low | Low |

Table 2: Feature descriptions of the top 10 states with the highest values of shaped-PDIS potential function in ICU-Sepsis. Green indicates good attributes. We observe these states have normal-high GCS, Normal BP, High SpO2 levels, Normal FiO2, normal White Blood Cell (WBC) count, normal HCO3 levels and normal partial pressure of arterial carbon dioxide (PaCO2).

| State Feature | $\phi(s)=1.73$ | $\phi(s)=1.29$ | $\phi(s)=1.14$ | $\phi(s)=1.03$ | $\phi(s)=1.01$ | $\phi(s)=0.90$ | $\phi(s)=0.88$ | $\phi(s)=0.86$ | $\phi(s)=0.85$ | $\phi(s)=0.75$ |
|---|---|---|---|---|---|---|---|---|---|---|
| age | Low | High | Medium | Low | Low | Medium | Low | Low | Low | Low |
| admissionweight | High | High | Medium | High | High | High | Medium | High | High | Medium |
| gcs | Low | Medium | Medium | Medium | Medium | High | Medium | Medium | Medium | Medium |
| hr | High | Low | Low | Low | High | Low | Low | High | High | High |
| sysbp | Low | Medium | Low | Medium | Medium | Medium | Low | Low | High | High |
| meanbp | Low | Medium | Low | Medium | Medium | Medium | Low | Low | High | High |
| diabp | Low | Medium | Low | Medium | Medium | Medium | Low | Low | High | High |
| rr | High | Medium | Medium | Medium | High | Medium | Low | High | High | Medium |
| temp_c | Medium | Low | Low | Low | High | Low | Low | High | High | High |
| fio2both | Medium | Medium | Medium | High | Medium | low | Medium | High | Medium | Medium |
| potassium | Medium | High | Low | High | Low | High | Medium | High | Low | High |
| sodium | High | Low | Low | Medium | High | Medium | Low | Low | High | Low |
| chloride | Low | Low | Low | Medium | Medium | Medium | Medium | Low | High | Low |
| glucose | High | High | Medium | Medium | High | High | Low | Low | Medium | Medium |
| magnesium | Medium | High | Low | High | High | High | Medium | Low | Low | High |
| calcium | Medium | High | Low | High | Medium | High | Medium | Low | Low | High |
| hb | High | Low | Low | Medium | Low | High | Low | High | Medium | Low |
| wbc_count | Medium | Medium | Low | Low | Low | Medium | Low | Medium | Medium | High |
| platelets_count | Low | High | Low | Low | Low | Medium | Low | Low | Low | High |
| ptt | High | Low | High | High | Low | Medium | High | High | Medium | High |
| pt | High | High | Medium | High | High | High | High | High | Low | Medium |
| arterial_ph | Low | Medium | High | Low | High | Medium | Low | Low | High | High |
| pao2 | Low | Low | Low | High | Medium | Medium | Medium | Low | Medium | Medium |
| paco2 | Low | High | Low | High | Medium | Medium | Low | Low | Medium | Medium |
| arterial_be | Low | High | Medium | Low | High | Medium | Low | Low | High | High |
| hco3 | Low | High | Medium | Medium | Medium | Medium | Low | Low | Medium | Medium |
| arterial_lactate | High | Low | Low | High | High | Low | High | High | Medium | Medium |
| sofa | High | High | High | High | High | Low | High | High | Low | High |
| sirs | High | Low | Low | Medium | High | Low | Low | High | High | High |
| shock_index | Medium | Medium | Medium | Medium | High | Medium | Medium | High | Medium | Medium |
| pao2_fio2 | Low | Low | Medium | High | Medium | High | Medium | Low | Medium | Medium |
| cumulated_balance_tev | High | Low | Medium | Medium | Low | Low | High | High | High | High |
| spo2 | Low | High | Medium | High | High | Medium | High | High | High | Medium |
| bun | High | High | High | High | High | Medium | High | High | Low | High |
| creatinine | High | High | High | High | High | Medium | High | High | Low | High |
| sgot | Medium | Medium | Low | Medium | Medium | Medium | High | High | High | Medium |
| sgpt | Medium | Medium | Low | Medium | Medium | High | Medium | High | High | Low |
| total-bili | High | High | Low | High | Medium | Medium | High | High | High | High |
| inr | High | High | Medium | High | High | High | High | High | Low | High |
| input-total-tev | High | Low | Low | Low | High | Low | High | High | High | High |
| input-4hourly-tev | High | Medium | Medium | Low | Medium | Low | High | High | High | Medium |
| output-total | Medium | Medium | Low | Low | High | Medium | Low | Low | High | Low |
| output-4hourly | Low | Medium | Low | Low | High | Low | Low | Low | High | Low |

