# OpenReview forum: "Reward Shaping Control Variates for Off-Policy Evaluation Under Sparse Rewards"
_ICLR.cc/2026/Conference — ICLR 2026 Conference Desk Rejected Submission_

### Official Review · Reviewer_m3u3 · 2025-10-30

**Soundness:** 3
**Presentation:** 3
**Contribution:** 3
**Rating:** 4
**Confidence:** 3

**Summary:**

This paper proposes an unbiased low-variance off-policy evaluation operator based on reward shaping control variates. The authors leverage the property of  state shaping potential functions do not alter the optimal policy to inject knowledge to sparse reward environments, where conventional methods like DR, MRDR can fail due to extreme sparsity. The authors prove their RSCV estimator strictly reduces variance with an optimal $\lambda^*$.

**Strengths:**

The paper nicely combines reward shaping and off-policy evaluation and clearly leverages the idea that potential based shaping does not shift the optimality of an MDP. The paper also surveys existing approaches like DR/MRDR on this problem to better position the proposed RSCV. Presentation is clear and understandable. Math is not deep but sufficient to illustrate the advantage of RSCV. Sepsis and cancer as testbeds are standard and the results seem convincing.

**Weaknesses:**

Disclaimer: I am not an expert in off-policy evaluation.

My concern is mainly technical.
 - Definition 1 defines potential based reward shaping $F(s, a, s') = \gamma \phi(s') - \phi(s)$. However, no action is involved in the RHS. Do the authors suggest an extended PBRS which is based on action? But in line 244 the authors write:
"$\Phi$ captures progress towards reward or risk of failure, so that the differences $\gamma\Phi(s_{t+1}) - \Phi(s_t)$ absorb predictable structure in the return". So it seems the potential is based solely on states. However, it confuses me why state-based potential could capture structure of reward that depends on actions?
- Line 183, I don't understand why the two sides equal under $W_t = \prod_k^t \frac{\pi_e(a_k|s_k)}{\pi_b(a_k|s_k)}$? Can you show more details?
- line 48 claims that marginalized estimators like DualDICE remain brittle due to the reward support. But isn't DICE methods estimate state marginals that do not depend on actions?
- line 160 claims that PBRS guarantees the evaluation policy's value is estimated consistently with the reward structure. But have the authors verified that holds (approximately) true with a parametrized $\Phi_{\beta}$? How do you guarantee the assumption $\mathbb{E}_{\pi_b}[C^{\Phi] = 0$ given a learned $\Phi_{\beta}$? If this holds only approximately, any insight on the difference it might bring to the result?

**Questions:**

please refer to weaknesses.

---

> ### Author Response · Authors · 2025-11-20
> **Author response to Reviewer m3u3: Part 1.**
>
> We sincerely thank the reviewer for the review and comments. We will sequentially address the questions.
>
> Q. Definition 1 defines potential based reward shaping . However, no action is involved in the RHS. Do the authors suggest an extended PBRS which is based on action? But in line 244 the authors write: " captures progress towards reward or risk of failure, so that the differences  absorb predictable structure in the return". So it seems the potential is based solely on states. However, it confuses me why state-based potential could capture the structure of reward that depends on actions?
>
> Response: Given an MDP environment, a single task can be represented by the two different kinds of reward functions, one dependent on just states, and one dependent on the entire (s,a,s’) transition. As an example, consider a 1D environment s1 - s2 - s3 where s3 is the goal, and the choices of actions are left and right. Now, these reward definitions are equivalent:
>
> Reward function 1: R(s3)=1, and R(s1)=R(s2)=0. You only get a reward when you arrive at state 3, and 0 otherwise. Note, there is no notion of actions in the reward function here. You learn your actions using your favorite policy gradient algorithm of your choice. Reward function 2: R(s2,action-right,s3)=1, R(s3,action-right,s3)=1, and 0 for all other transitions. Here, there is an explicit dependence on which action you take given a state to reveal your reward. Both definitions are correct, however just using states is simpler.
>
> This notion also applies to potential functions. Potential functions can just be dependent on states, can give you a scalar value and can directly guide whether the action leading upto the reward was good or bad. So, while actions aren’t explicitly inputs to the potential function, they can be evaluated using $\phi$ which can just depend on states.
>
> There also exists action-based potential functions, wherein $\phi(s,a)$ takes both the state and the action as input, [1,2], but it’s easier to maintain the fidelity of the task and not change the optimality by making the potential function just depend on states. We cover more on this in your “consistency question” later in the rebuttal. Additionally, $\phi(s,a)$ requires more samples to be learnt and it’s sample efficiency is comparable to actually learning a value estimate as in DR.
>
> Q. Line 183, I don't understand why the two sides are equal under $W_t$? Can you show more details?
>
> Response: Yes. It’s a simple probability change of measure trick, popularly known as Radon-nikodym derivatives.
>
> $$
> \mathbb{E}_{\pi_b}
> $$
>
> $$
> \left[\gamma^t
> \left(\prod_{k=0}^T \frac{\pi_e(a_k\mid s_k)}{\pi_b(a_k\mid s_k)}\right)
> \big(\gamma\Phi(s_{t+1})-\Phi(s_t)\big)\right]
> \tag{a}
> $$
>
> $$
> = \frac{1}{N}\sum_{n=1}^N
> \left(\prod_{k=0}^T \pi_b(a_k\mid s_k)\right)
> \left[\gamma^t
> \left(\prod_{k=0}^T \frac{\pi_e(a_k\mid s_k)}{\pi_b(a_k\mid s_k)}\right)
> \big(\gamma\Phi(s_{t+1})-\Phi(s_t)\big)\right]
> \tag{b}
> $$
>
> $$
> = \frac{1}{N}\sum_{n=1}^N
> \left[\gamma^t
> \left(\prod_{k=0}^T \pi_e(a_k\mid s_k)\right)
> \big(\gamma\Phi(s_{t+1})-\Phi(s_t)\big)\right]
> \tag{c}
> $$
>
> $$
> \mathbb{E}_{\pi_e}
> $$
>
> $$
> \left[\gamma^t \big(\gamma\Phi(s_{t+1})-\Phi(s_t)\big)\right]
> \tag{d}
> $$
>
> Explanation of steps:
>
> (a) Expanding the importance sampling ratio W_t
>
> (b) Expanding the definition of expectation under behavior policy.
>
> (c) $\prod_{t=0}^T(\pi_b(a_t|s_t))$ is the probability of the trajectory
> This probability cancels with the denominator of W_t
>
> (d) Pulling $\prod_{t=0}^T(\pi_e(a_t|s_t))$ outside the summation and re-writing it as expectation under evaluation policy. This completes the small lemma.
>
> [1] Potential-Based Advice for Stochastic Policy Learning Xiao et al.
>
> [2] Principled Methods for Advising Reinforcement Learning Agents Wiewiora et al.

---

> ### Author Response · Authors · 2025-11-20
> **Author response to Reviewer m3u3: Part 2.**
>
> Q. Line 48 claims that marginalized estimators like DualDICE remain brittle due to the reward support. But isn't DICE methods estimate state marginals that do not depend on actions?
>
> Response:
>
> Marginalised estimators can be defined based on both, just the state marginals as well as the state-action marginals [1]. However, it is more popular to use state-action marginals, as the policy evaluation is directly engrained into the density ratios via the actions. A typical Marginalised estimator is of the form:
>
> $$\hat{V}^{\pi_e}=\frac{1}{N}\sum_{n=1}^N \frac{d_{\pi_e}(s,a)}{d_{\pi_b}(s,a)}r(s,a)$$
>
> While DualDICE algorithms focus on estimating the ratio \frac{d_{\pi_e}(s,a)}{d_{\pi_b}(s,a)}, the denser the reward information r(s,a) across various state-action pairs, the better the consistency of the estimator. Unfortunately, under sparse reward scenarios, the MIS estimator isn’t in a position to take advantage of the state or state-action density ratios. Thus, shaping algorithms can potentially benefit even estimators from the MIS family by densifying the reward signals.
>
> Marginalised estimators, however, have their caveats. Marginalised estimators were roped in to counter the curse-of-horizon problem that you typically encounter in the IPS ratios due to very high IPS values. However, it depends on estimating $d_{\pi_e}(s,a)$, a quantity not known beforehand and needs to be estimated. DICE algorithms attempt to solve this. Problem with DICE algorithms: They solve a minimax optimization problem which converges only if the loss landscape is convex enough. [2,3]. Unfortunately, most RL environments don’t fall in this regime, and even slight errors of density ratio estimations adds large bias into the estimator. [4,5,6].
>
> We conducted an ablation experiment (Figure 6) to support this. We considered DualDICE, GenDICE and GradientDICE as our DICE algorithms and compared them against shaped-PDIS, shaped-DR and shaped-MRDR. We observe, DualDICE and GenDICE completely collapse due to lack of reward signals (It has been theoretically proven DualDICE algorithm is unstable with large discount factors [7]). GradientDICE algorithm is stabler, has a low variance, but due to inaccuracy in estimating the density ratio, they incorporate a huge bias, and consequently MSE. Even the simplest shaped estimator (shaped-PDIS) outperforms all the DICE estimators despite incorporating the curse-of-horizon problem, indicating the strength of the potential functions and the algorithm to improve the OPE estimators.
>
> Q. Line 160 claims that PBRS guarantees the evaluation policy's value is estimated consistently with the reward structure. But have the authors verified that holds (approximately) true with a parametrised $\phi$? How do you guarantee the assumption $\mathbb{E}^{\pi_b}[C^{\Phi}] = \mathbb{E}^{\pi_e}[C^{\Phi}] = 0$? If this holds only approximately, any insight on the difference it might bring to the result?
>
> There are 3 different timesteps under which the potential function $\phi$ is defined. One, at the final timestep T, we define $\phi(s_T)=0$, as we typically know the reward outcome at the final timestep. Second, at the first timestep, we assume $\mathbb{E}^{\pi_b}[C^{\Phi}] = \mathbb{E}^{\pi_e}[C^{\Phi}] = 0$, not because we know the reward, but generally in practice we have a notion of what’s our initial distribution. Typically, in practice, we have knowledge about those states, we don’t actively “shape” it in forms of rewards. Then, comes steps 1 to t-1, which is where we try to learn the shaping function $\phi$ which actively serves as a reward signal while evaluation. In line 160, by ‘This invariance property makes PBRS especially appealing for OPE: it allows us to introduce additional signal into sparse-reward environments while guaranteeing that the evaluation policy’s value is estimated consistently with the original reward structure’, we mean that we learn shaping potentials that depend only on states. As a result there is no contamination to the policy and the environmental setup, thus the fidelity of the task is maintained.
>
> [1] DualDICE: Behavior-Agnostic Estimation of Discounted Stationary Distribution Corrections Nachum et al.
>
> [2] GradientDICE: Rethinking Generalized Offline Estimation of Stationary Values Zhang et al.
>
> [3] Train simultaneously, generalize better: Stability of gradient-based minimax learners Farnia et al.
>
> [4] Understanding the Curse of Horizon in Off-Policy Evaluation via Conditional Importance Sampling Liu et al.
>
> [5] A Deep Reinforcement Learning Approach to Marginalized Importance Sampling with the Successor Representation Fujimoto et al.
>
> [6] Practical MIS with Successor Representation Fujimoto et al.
>
> [7] GENDICE: Generalised Offline Estimation of stationary values. Zhang et al.

---

> > ### Author Response · Authors · 2025-11-25
> > **Any further clarifications that can help with the understanding of the paper?**
> >
> > Dear Reviewer m3u3,
> >
> > We once again thank you for your comments. We have responded to all your questions sequentially and incorporated additional analysis in the revised version of the paper. Is there anything else you would like to see in the revised version or answer via rebuttals that can help further enhance the paper and increase the score?

---

> > ### Comment · Reviewer_m3u3 · 2025-11-26
> >
> > Thank you for the detailed response.
> >
> > I believe ICLR allows for expanding to 10 pages during rebuttal. I will raise my score if the authors can incorporate these changes to the updated draft.

---

> > > ### Author Response · Authors · 2025-11-26
> > > **Summary of our changes in the revised version of the paper**
> > >
> > > Dear Reviewer m3u3,
> > >
> > > Thank you for letting us know we have an extra page to add the additional analysis and experiments. We have made the updates, and the summary of the requested changes are as follows:
> > >
> > > 1. Lines 158-161, Page 3: Clarification of potential functions not changing consistency with change in the reward function.
> > > 2. Lines 184-190, Page 4: Clarification on the assumptions surrounding C^\Phi. (Response to your final rebuttal question).
> > > 3. Lines 478-515, Pages 9-10: Additional ablation experiments with Marginalised Importance Sampling baselines over the ICU-Sepsis example. Also covered is the discussion surrounding why shaped estimators outperform MIS estimators due to sparse reward values and brittleness in the density ratios.
> > > 4. Lines 449-464, Page 8: Cancer simulator results. We moved the picture from the Appendix to the main section for completeness.
> > > 5. Lines 648-670, Page 13, Appendix B: Added intuition behind the shaping function from the lens of OPE. (Response to your 1st rebuttal question).
> > > 6. Lines 675-701, Page 13, Appendix C: Extended Proof of Lemma 1 (Response to your 2nd rebuttal question).
> > >
> > > Please let us know if you need any further clarification. Thank you for helping us improve the paper!
> > >
> > > Kind Regards,
> > > Authors

---

### Official Review · Reviewer_W8Fe · 2025-10-31

**Soundness:** 2
**Presentation:** 2
**Contribution:** 3
**Rating:** 2
**Confidence:** 4

**Summary:**

The paper tackles the challenge of doing off-policy evaluation for sparse reward tasks by proposing reward-shaping control variates (RSCV). The method uses a potential-based reward shaping technique to maintain the optimal policy under the shaped MDP. It defines a learnable additional random variable which has a zero mean under the behavior-policy distribution, and learns the random variable by a potential network for variance reduction. The work is evaluated on medical treatment benchmarks.

**Strengths:**

The method is supported by theoretical proof, showing the modification added to the reward does not change the optimal action distribution (Theorem 1), and remains unbiased (Theorem 2). Theorem 3 and Corollary 1 show the bound of variance.

The paper discusses the difference between the proposed method and related works, DR and MRDR (Section 4.3). The discussion highlights that RSCV mainly relies on learning the potential function, while related works focus on learning the action value estimation.

The paper performs a sanity check in a clear tabular environment. A simple tabular environment is helpful in providing us with a clear demonstration of RSCV’s effectiveness with an increasing sample size.

The paper considers multiple evaluation metrics, including the bias, variance, mean-squared error, and effective sample size, supporting the theoretical result regarding the bias and variance.

**Weaknesses:**

The method is limited to discrete action space tasks. For datasets with continuous action spaces, the paper discretizes the space (indicated in F.2). Discretizing values introduces information loss because different actions may be mapped to the same discrete representation. In addition, the choice of discretization parameters, such as the number of bins, can significantly affect performance and stability.

Fitted Q evaluation (FQE) is used as the action value estimation in DR as a baseline, however, it is not a strong baseline choice especially when the dataset coverage is limited. FQE does not properly handle the out-of-distribution action sample in bootstrapping, which can cause inaccurate value estimates even after many training iterations.

The paper discusses marginalized estimators such as DualDICE and GenDICE in the related work section, but does not include them in the empirical comparison. It may be worth checking their performance as well.

Experimental validation is conducted on only three datasets, one of which serves as a sanity check. Evaluating the method on a broader range of benchmarks would make the empirical results more convincing.

The method introduces a weighting parameter ($\lambda$) for the learnable random variable term, but lacks the sensitivity analysis on it. It would be good to discuss how the weight affects performance.

**Questions:**

I have the following 2 questions:

1. To optimize the model, the paper set the training process to use k-fold validation. K-fold is an effective way to prevent overfitting, but at the cost of longer computational time and larger computational resources. Could the author please further explain the computational overhead for introducing k-fold here? With a large dataset, is it recommended to further reduce the K? According to the current experiment, is the difference large when running with different K’s?

2. Could the author please explain what the error bars in the figures represent? Is it the standard deviation?

I would be happy to discuss further if I have misunderstood any part of the method or experimental setup.

---

> ### Author Response · Authors · 2025-11-20
> **Author response to Reviewer W8Fe: Part 1.**
>
> We sincerely thank the reviewer for the review and comments. We will sequentially address the questions.
>
> Q. The method is limited to discrete action space tasks. For datasets with continuous action spaces, the paper discretizes the space (indicated in F.2). Discretizing values introduces information loss because different actions may be mapped to the same discrete representation. In addition, the choice of discretization parameters, such as the number of bins, can significantly affect performance and stability.
>
> Response: We agree with the potential information loss that arises due to discretizing values. Point taken. This however, affects all estimators equally, and is a problem with the existing estimators even under dense reward settings. While our experiments address discrete states and actions, we impose no limitations on the input domain of our shaping functions, and thus, our proposed algorithm can be easily extended to the continuous state and actions as well. This is supported theoretically in the confidence bounds (Page 15, Appendix D1) where-in the lower bound of variance is independent of the cardinality of both states and actions. The ICU-Sepsis example is a simulator based out of the real world MIMIC-IV dataset, which contains 716 states, with each state being 47 dimensional continuous valued features with 25 actions. While discrete, this problem is much higher in complexity as compared to many of the toy Atari and Mujoco tasks as they are environments with dense reward signals. The dense reward signals makes it easier to obtain policy overlaps while performing IS. We have added the discretization parameters in Appendix F, and for further info, we point you to the original ICU-Sepsis environment construction [1].
>
> Q. Fitted Q evaluation (FQE) is used as the action value estimation in DR as a baseline, however, it is not a strong baseline choice especially when the dataset coverage is limited. FQE does not properly handle the out-of-distribution action sample in bootstrapping, which can cause inaccurate value estimates even after many training iterations.
>
> Response: FQE is quite standard in the OPE literature as a benchmark, and we wanted to capture the fact that shaped-DR performs well inspite of taking a poor model-based FQE as an estimate. This is because DR is designed such that even if one of the estimators, i.e. the model-based FQE or the Importance Sampling has lower bias, the overall estimator is lower biased. This holds for the shaping variants as well, as seen in the optimal performance of shaped DR and shaped MRDR.
> We added three other baselines (Page 18, Figure 6), two from the Marginalised IS family (response to next question) and Magic-FQE, one of the superior FQE estimators. MAGIC [2] is a hybrid method that does not rely solely on the model (FQE) nor solely on Importance Sampling (IS). Instead, it calculates a set of Doubly Robust (DR) estimates over varying partial trajectory lengths and blends them. We observe, our shaped DR and MRDR estimators have similar bias but just slightly higher variance than Magic-FQE, even on low sample regimes. With better model-based estimators like Magic-FQE incorporated into shaped-DR and shaped-MRDR, the shaping performance can potentially further improve.
>
> Q. The paper discusses marginalized estimators such as DualDICE and GenDICE in the related work section, but does not include them in the empirical comparison. It may be worth checking their performance as well.
>
> Response: We conducted an ablation experiment to support this (Figure 6).  We considered DualDICE, GenDICE and GradientDICE as our DICE algorithms and compare them against shaped-PDIS, shaped-DR and shaped-MRDR. We observe DualDICE and GenDICE completely collapse due to lack of reward signals (It has been theoretically proven DualDICE algorithm is unstable with large discount factors). GradientDICE  is more stable, has a low variance, but due to inaccuracy in estimating the density ratio, has a huge bias, and consequently high MSE. Even the simplest shaped estimator (shaped-PDIS) outperforms all the DICE estimators in spite of the curse-of-horizon problem, indicating the strength of the potential functions and the algorithm to improve the OPE estimators.
>
> [1] ICU-Sepsis: A Benchmark MDP Built from Real Medical Data Chowdhary et al.
> [2] Data-Efficient Off-Policy Policy Evaluation for Reinforcement Learning Philip S. Thomas, Emma Brunskill

---

> ### Author Response · Authors · 2025-11-20
> **Author response to Reviewer W8Fe: Part 2.**
>
> Q. Experimental validation is conducted on only three datasets, one of which serves as a sanity check. Evaluating the method on a broader range of benchmarks would make the empirical results more convincing.
>
> Response: The most popular environments in Mujoco and Atari are mostly based on dense rewards, hence they can’t be directly used in our problem setup. As sparsity of rewards is typically encountered in healthcare, we used Cancer simulator and ICU-Sepsis as our datasets. Could you suggest any datasets? While it’s tight in terms of time, we will try to run any additional experiments if time permits and ACs, other reviewers agree.
>
>
> Q. The method introduces a weighting parameter $\lambda$ for the learnable random variable term, but lacks the sensitivity analysis on it. It would be good to discuss how the weight affects performance.
>
> Response: The $\lambda$ term directly controls the impact of the shaping function on the final estimator. We derived $\lambda = -Cov(Y,C)/Var(C)$ to be the estimator that has the minimum variance. Depending on the values of $\lambda$, one can produce different estimators. A specific instance of $\lambda=-1$ actually recovers the Doubly Robust estimator, as proven in [1]. This estimator, however isn’t guaranteed to reduce variance over a normal estimator without shaped reward. This also allows for shaped estimators optimised for quantities other than variance, as an example, if the on-policy value function is hypothetically known, one can learn an appropriate $\lambda$ to get the unbiased or the minimum MSE estimator. Additionally, for sensitivity analysis, we provide comparisons when the control variate is contaminated with noise in Page 9, Figure 4, wherein we observe shaped-estimators are actually more robust than the traditional estimators. Additionally, we also compare the estimator with the reward density (% of state-action pairs assigned a reward) in Page 17, Figure 5.
>
> Q. To optimize the model, the paper set the training process to use k-fold validation. K-fold is an effective way to prevent overfitting, but at the cost of longer computational time and larger computational resources. Could the author please further explain the computational overhead for introducing k-fold here? With a large dataset, is it recommended to further reduce the K? According to the current experiment, is the difference large when running with different K’s?
>
> Response: You are right to point out the overhead of the algorithm. The largest overhead and potentially the bottleneck of the algorithm lies in estimating the inverse of the covariance matrix in Algorithm 1, with the dependence increasing cubically with the number of samples. This cost is lower for the smaller datasets, hence we can use large values of K. With a larger dataset, the generalisation over the visited state-action pairs improves and there are higher overlaps between evaluation and behavior policies, so values of K can be reduced significantly, and theoretically K tends to 1 as the number of samples grows to infinity. In our current experiments, across all datasets, we observe relatively more stable estimates of the value function (N>=1000 for ICU-Sepsis), which is supported by very low standard deviations across all metrics (The answer to your next question). To improve computational overhead, a practitioner can replace the computation of the inverse of the covariance matrix with conjugate gradient methods.
>
> Q. Could the author please explain what the error bars in the figures represent? Is it the standard deviation?
>
> Response: Yes, the error bars represent the standard deviations. We conduct each OPE experiment (both the shaped variants and the baselines) over 10 seeds. The lower the error-bar interval, more consistent the estimator performance is.
>
> [1] Shaping Control Variates for Off-Policy Evaluation Parbhoo et al.

---

> > ### Comment · Reviewer_W8Fe · 2025-11-21
> >
> > I sincerely thank the authors for their thorough responses and the additional experiments. I think my questions have been well addressed, particularly regarding the choice of baselines and dataset selection, though I remain unconvinced by the comparison between the difficulty of sparse-reward discrete control tasks and dense-reward continuous control tasks, which I believe would require a more careful definition and quantification. I now have a better understanding of the experimental choices overall. Based on these clarifications, I am happy to increase my score.

---

> > > ### Author Response · Authors · 2025-11-25
> > >
> > > Thank you, reviewer W8Fe for your timely response to our rebuttal and constructive comments that helped improving our paper.

---

### Official Review · Reviewer_nuKr · 2025-11-02

**Soundness:** 3
**Presentation:** 3
**Contribution:** 3
**Rating:** 6
**Confidence:** 2

**Summary:**

This paper addresses the critical failure of standard off-policy evaluation (OPE) estimators, such as Importance Sampling (IS) and Doubly Robust (DR), which suffer from prohibitively high variance in sparse-reward environments. The authors introduce Reward-Shaping Control Variates (RSCV), a new class of unbiased estimators. The core idea is to leverage policy-invariant potential-based reward shaping (PBRS) to construct a new, provably zero-mean control variate. The authors provide a practical algorithm to learn the potential function $\Phi$ and the optimal weight $\lambda$ directly from data by maximizing variance reduction. Experiments on a chain MDP, a cancer simulator, and the ICU-Sepsis benchmark show that this method reduces variance and Mean Squared Error (MSE) significantly compared to standard baselines.

**Strengths:**

- The method directly targets a well-known and significant limitation of OPE, high variance in sparse-reward settings, which is a common feature of real-world applications such as healthcare.
- The experimental results demonstrate orders of magnitude improvement in variance and MSE over PDIS, DR, and MRDR in sparse-reward tasks. The method is also shown to be more robust to reward noise.

**Weaknesses:**

- The method's success depends on learning a good potential function $\Phi$. The experiments are on tabular or low-dimensional (4-47 features) state spaces. How well does the learning algorithm for $\Phi$ (Algorithm 1) scale to high-dimensional problems (e.g., images)?
- There is a contradiction in the ICU-Sepsis setup. Section 5.3 states $\pi_b$ is a generated PPO policy, while Appendix F.2 says it's the empirical clinician policy (Line 895). Which was it?

**Questions:**

See Weaknesses.

---

> ### Author Response · Authors · 2025-11-20
> **Author response to Reviewer nuKr**
>
> We sincerely thank the reviewer nuKr for the review and comments. We will sequentially address the questions.
>
> Q. The method's success depends on learning a good potential function . The experiments are on tabular or low-dimensional (4-47 features) state spaces. How well does the learning algorithm for  (Algorithm 1) scale to high-dimensional problems (e.g., images)?
>
> Response:
> With an increase in the complexity of the task, either by scaling to high-dimensional states or actions, the performance of all existing OPE estimators further deteriorates. With sparser rewards, existing OPE estimators of the Importance Sampling family (e.g. IS, MIS, PDIS, DR) are prone to even higher variance, while direct and model-based methods suffer from higher bias. This problem persists regardless of the dimensionality of the states and actions. Our algorithm provides an opportunity to improve the estimators by densifying the rewards in the form of potential functions. Furthermore, as we make no assumptions on the cardinality of the states and actions, our algorithm is general enough to function even in continuous state-action setups. This is supported theoretically in the confidence bounds (Appendix D1) where-in the lower bound of variance is independent of the cardinality of both states and actions.
>
> Q. There is a contradiction in the ICU-Sepsis setup. Section 5.3 states is a generated PPO policy, while Appendix F.2 says it's the empirical clinician policy (Line 895). Which was it?
>
> Response:
> The behavior and evaluation policies are generated using PPO over 1M epochs, with the behavior policy being the model parameters of the actor at epoch 250k and evaluation policy being the model parameters of the actor at epoch 1M. The authors in [1] constructed an ICU-Sepsis environment from the real world MIMIC data, wherein the authors used the empirical clinician actions to guide the model training process. We are using the environment directly, upon which we run PPO to get our policies. We have corrected this in the updated version of the paper.
>
> [1] ICU-Sepsis: A Benchmark MDP Built from Real Medical Data. Kartik Choudhary, Dhawal Gupta, Philip S. Thomas

---

> > ### Author Response · Authors · 2025-11-25
> > **Any further clarifications that can help with the understanding of the paper?**
> >
> > Dear Reviewer nuKr,
> >
> > We once again thank you for your comments. We have responded to all your questions sequentially and incorporated additional analysis in the revised version of the paper. Is there anything else you would like to see in the revised version or answer via rebuttals that can help further enhance the paper and increase the score?

---

### Note · Program_Chairs · 2026-01-17
**Submission Desk Rejected by Program Chairs**

The following references in this submission do not refer to real documents and/or have major errors in bibliographic information:

 Masatoshi Uehara, Ofir Nachum, and Nan Jiang. Review of off-policy evaluation in reinforcement learning. arXiv preprint arXiv:2204.05440, 2022.